# VLBiMan: Vision-Language Anchored One-Shot Demonstration Enables Generalizable Bimanual Robotic Manipulation

**Huayi Zhou**[1]     **Kui Jia**[1,2,*]
[1]The Chinese University of Hong Kong, Shenzhen     [2]DexForce, Shenzhen
`zhouhuayi@cuhk.edu.cn; kuijia@cuhk.edu.cn`
`https://hnuzhy.github.io/projects/VLBiMan`

## Abstract

Achieving generalizable bimanual manipulation requires systems that can learn efficiently from minimal human input while adapting to real-world uncertainties and diverse embodiments. Existing approaches face a dilemma: imitation policy learning demands extensive demonstrations to cover task variations, while modular methods often lack flexibility in dynamic scenes. We introduce VLBiMan, a framework that derives reusable skills from a single human example through task-aware decomposition, preserving invariant primitives as anchors while dynamically adapting adjustable components via vision-language grounding. This adaptation mechanism resolves scene ambiguities caused by background changes, object repositioning, or visual clutter without policy retraining, leveraging semantic parsing and geometric feasibility constraints. Moreover, the system inherits human-like hybrid control capabilities, enabling mixed synchronous and asynchronous use of both arms. Extensive experiments validate VLBiMan across tool-use and multi-object tasks, demonstrating: (1) a drastic reduction in demonstration requirements compared to imitation baselines, (2) compositional generalization through atomic skill splicing for long-horizon tasks, (3) robustness to novel but semantically similar objects and external disturbances, and (4) strong cross-embodiment transfer, showing that skills learned from human demonstrations can be instantiated on different robotic platforms without retraining. By bridging human priors with vision-language anchored adaptation, our work takes a step toward practical and versatile dual-arm manipulation in unstructured settings.

## 1 Introduction

Recent years have witnessed rapid progress in embodied robotic manipulation, particularly under the paradigm of visuomotor imitation learning through large-scale teleoperated demonstrations Fang et al. (2024a); Khazatsky et al. (2024); O'Neill et al. (2024); Bu et al. (2025). By collecting thousands of real-world samples for each task and object setting, Vision-Language-Action (VLA) models Team et al. (2024); Kim et al. (2024); Lin et al. (2025) are trained to directly map raw sensory inputs to motor commands. This end-to-end approach avoids explicitly modeling task- or object-specific priors (even for challenging cases involving deformable or articulated objects), by embedding such complexities into high-dimensional latent representations. Such strategies are especially compatible with high-DoF collaborative scenarios like bimanual manipulation, enabling impressive performance on long-horizon tasks, as demonstrated by works such as ALOHA series Zhao et al. (2023a); Fu et al. (2024); Aldaco et al. (2024); Zhao et al. (2024), RDT-1B Liu et al. (2025a), $\pi_0$ Black et al. (2024), and FAST Pertsch et al. (2025). However, this line of research is **bottlenecked** by its reliance on large-scale data collection and retraining cycles: adapting to new objects or tasks typically demands a full demonstration pipeline and model retraining, hindering scalability in open-world settings with unbounded task-object combinations and robot types.

To alleviate this, recent efforts have embraced modularized VLA pipelines that leverage the generalization capabilities of pre-trained LLMs Achiam et al. (2023) and VLMs Radford et al. (2021); Xiao et al. (2024). These models are repurposed to handle perception and semantic grounding, while downstream motion execution is delegated to either optimization-based controllers or pretrained visuomotor modules such as atomic skills or diffusion policies Chi et al. (2023); Ze et al.

---

*The corresponding author.

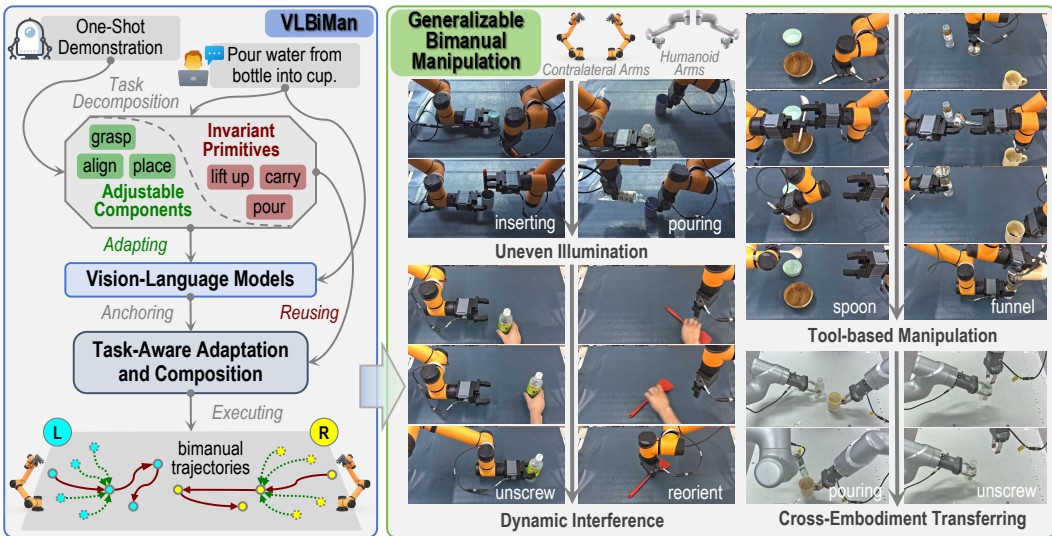

Figure 1: *Left*: Taking pouring water as an example, we sketch the entire process of VLBiMan based on the one-shot demonstration. *Right*: VLBiMan can achieve generalizable bimanual manipulation on a variety of complex contact-rich tasks without retraining, robustly coping with diverse scenarios.

(2024); Yang et al. (2024). Reinforcement learning in simulation also serves as a strategy for learning skill-specific controllers Xie et al. (2020); Chen et al. (2022); Yuan et al. (2024b). This modular design allows robotic agents to inherit part of the generalization capability from foundation models, while maintaining flexibility and interpretability. A common practice in these pipelines is to define generalizable representations (*e.g.*, keypoints, affordances and correspondences), as structured anchors between perception and control. For instance, ReKep Huang et al. (2024b) plans robot motion by anchoring on multiple predicted relation points, MOKA Fang et al. (2024b) extracts fine-grained functional regions via multi-modal visual question answering, and RobotPoint Yuan et al. (2024a) identifies object-centric task-relevant point clusters. Such approaches demonstrate that keypoint-affordance abstractions are effective for transferring behavior across objects, viewpoints, or instances, and have become a cornerstone of generalizable manipulation.

Building on this insight, we propose **VLBiMan** for one-shot bimanual manipulation that leverages vision-language anchoring without retraining. Our approach also relies on object-centric representation points, but rather than predicting them via learned networks, we utilize VLMs to perform stable and robust object segmentation, followed by two heuristic strategies for anchor selection: geometric center of masks and plane-contact points. These anchors, though reminiscent of affordances, are far more controllable and lightweight. Unlike prior zero-shot methods Huang et al. (2024b) that require fragile prompt engineering and suffer from unreliable trajectory execution, our framework is demonstration-conditioned: we structure the action plan based on a one-shot, fine-labeled demonstration, then adapt it using language-grounded object anchors and motion optimization techniques. This enables robust execution on complex bimanual tasks while reusing invariant sub-skills.

Our methodology unfolds in three stages: (1) *Task-Aware Bimanual Decomposition*, which splits the one-shot demonstration into semantically meaningful left/right arm primitives with inter-arm dependencies; (2) *Vision-Language Anchored Adaptation*, which grounds the invariant motion primitives onto new scenes by aligning demonstration anchors with newly segmented objects via VLMs; (3) *Autonomous Trajectory Composition*, which composes new robot trajectories through kinematics-aware blending of adapted sub-skills, ensuring smooth coordination under scene variations. The related illustrations can be glimpsed in Fig. 1 and Fig. 2. VLBiMan actually is inspired by a key principle: **what to achieve matters more than how to execute it**. For instance, rather than mimicking the exact poses or insignificant diversities involved in pouring water, our approach focuses on capturing and re-instantiating the relative spatial relationship between the cup and bottle, emphasizing coordination rather than absolute motion. We validate VLBiMan across ten diverse bimanual tasks (including six basic bimanual skills, two long-horizon tasks consisting of skill combinations, and two multi-stage tool-use tasks), demonstrating superior generalization and minimal engineering overhead compared to prior strong baseline methods.

To summarize, our contributions are as follows: (**i**) We propose VLBiMan, a novel framework that enables generalizable bimanual manipulation through one-shot demonstration and vision-language anchoring, without retraining. (**ii**) We introduce a task-aware motion decomposition and adaptation mechanism, which reuses invariant sub-skills via object-centric anchors from VLMs and supports cross-embodiment transfer from human demonstrations to different robotic embodiments. (**iii**) We validate VLBiMan on ten diverse bimanual tasks, showing superior generalization, sample efficiency, and robustness compared to strong baselines.

## 2 RELATED WORKS

**Generalizable Representations for Manipulation.** Traditional robotic manipulation often relied on structured representations built upon strong priors Kaelbling & Lozano-Pérez (2013); Dantam et al. (2018); Migimatsu & Bohg (2020); Tyree et al. (2022), such as object geometry or rigid-body assumptions, typically via estimating 6D poses or manually specifying grasp configurations. They are hard to scale in unstructured environments. With the rise of data-driven techniques, more flexible representations have emerged, including keypoints Papagiannis et al. (2024); Gao et al. (2024); Wen et al. (2024b); Grannen et al. (2021), affordances Ju et al. (2024); Nasiriany et al. (2024); Zhao et al. (2023b), dynamic flow fields Colomé & Torras (2018); Weng et al. (2022), and invariant object-centric correspondences Ko et al. (2024); Zhang & Boularias (2024); Zhang et al. (2023). Some works further leverage human demonstrations to retarget 3D hand trajectories to robots Chen et al. (2024a); Li et al. (2024); Kerr et al. (2024); Chen et al. (2024b). However, these approaches often depend on private datasets, retraining, or complex retargeting pipelines, limiting scalability. In contrast, our method essentially anchors adaptation to object representative points without retraining, achieving greater efficiency and generality.

**Efficient Bimanual Robotic Manipulation.** Recent advances in bimanual manipulation have showcased the power of large Vision-Language-Action (VLA) models Black et al. (2024); Liu et al. (2025a); Pertsch et al. (2025) trained on extensive teleoperated demonstrations Fang et al. (2024a); Khazatsky et al. (2024); O'Neill et al. (2024); Bu et al. (2025). However, these approaches are highly suspected of lacking efficiency, as scaling to unseen objects or tasks often requires re-collecting and retraining. Alternative efforts explore leveraging large-scale Internet Ponimatkin et al. (2025); Ye et al. (2025); Bharadhwaj et al. (2024) or egocentric human-hand videos Zhan et al. (2024); Liu et al. (2024b); Grauman et al. (2024); Zhao et al. (2025); Kareer et al. (2024), yet the embodiment gap between human and robot limits direct usability. Some methods improve sample efficiency by learning visuomotor policies Chi et al. (2023); Ze et al. (2024) from a small set of real-world robot data, but their generalization remains limited. While one-shot imitation learning Wen et al. (2022); Bahety et al. (2024); Zhou et al. (2025); Wang & Johns (2025); Mao et al. (2023); Liu et al. (2025b); Biza et al. (2023); Zhou & Jia (2026); Zhou et al. (2026) reduces data demands, the high-dimensional action space and coordination complexity of bimanual control hinder learning efficiency. In contrast, VLBiMan achieves efficient adaptation from a single bimanual demonstration by leveraging VLMs to handle novel variations, while reusing decomposed task-invariant atomic skills. These lead to both data and computational efficiency.

**Large Foundation Models for Robotics.** Integrating LLMs and VLMs into robotics is a prominent trend to enable generalizable agents Ma et al. (2025); Huang et al. (2025); Fang et al. (2025); Feng et al. (2025). LLMs are utilized for high-level task understanding and planning, such as decomposing instructions into executable subtasks or generating scripts Liang et al. (2023); Singh et al. (2023); Szot et al. (2024); Huang et al. (2024a). Meanwhile, VLMs facilitate visually grounded perception through semantic prompts, enabling object-level detection and segmentation. For fine-grained tasks, Visual Foundation Models (VFMs) Oquab et al. (2024); Ravi et al. (2025) are further employed to find keypoints Papagiannis et al. (2024); Gao et al. (2024); Wen et al. (2024b) or dense correspondences Ko et al. (2024); Zhang & Boularias (2024). Recent efforts like ReKep Huang et al. (2024b), MOKA Fang et al. (2024b), RobotPoint Yuan et al. (2024a), and RAM Kuang et al. (2024) combine LLMs and VLMs into modular pipelines that follow the *perceive-understand-plan-act* paradigm to achieve zero-shot generalization. These approaches often rely on engineered prompts and ambiguous intermediate representations (*e.g.*, region of interest or keypoint clusters) requiring additional post-processing. In contrast, VLBiMan avoids LLM-based instruction parsing and task decomposition, which are brittle and labor-intensive. Instead, we build on one-shot demonstrations with precise action labels, using VLMs to extract semantically grounded action structures that are adaptively composed and reused, enabling efficient and scalable bimanual manipulation.

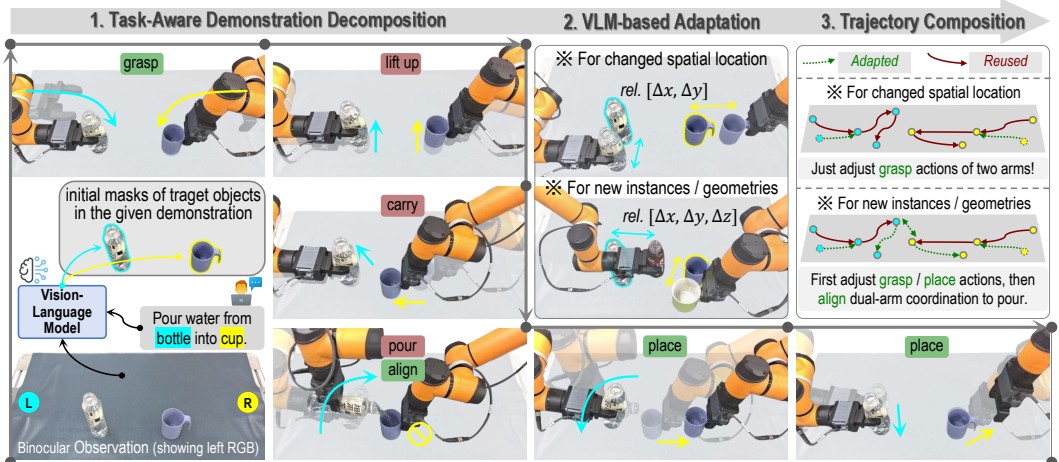

Figure 2: Framework of **V**ision-**L**anguage Anchored **Bi**manual **Man**ipulation (**VLBiMan**). Taking the pouring water as an example, the paradigm consists of three stages (*e.g.*, decomposition, adaptation, and composition) based on a given demonstration. VLBiMan can achieve generalization of unseen spatial placements and category-level new instances under the same task.

## 3  METHODOLOGY

This section introduces the full pipeline of **VLBiMan** (Fig. 2), which enables generalizable bimanual manipulation via vision-language anchored one-shot demonstration. Firstly, we present preliminaries, where we formalize the problem and describe the input-output configuration. Then, we explain three key components: (1) Task-Aware Bimanual Decomposition in Sec. 3.1, which extracts reusable atomic skills through structured trajectory segmentation; (2) Vision-Language Anchored Adaptation in Sec. 3.2, which adapts to new object instances or configurations with vision-language models; and (3) Autonomous Trajectory Composition in Sec. 3.3, which composes and optimizes executable dual-arm motion plans under physical and semantic constraints.

**Preliminaries**. Given a concise textual description of a bimanual manipulation task, together with an one-shot demonstration in a canonical scene, we aim to synthesize executable dual-arm trajectories through modular decomposition and adaptation in new scenes, where objects may be relocated or replaced by category-level variants. Formally, let $\mathcal{T}$ denote the task description and $\mathcal{D} = \{(\mathcal{O}_t, \mathcal{A}_t)\}_{t=1}^{T}$ represent the demonstration, where $\mathcal{O}_t$ is the multimodal observation (*e.g.*, visual frame, 6-DoF end-effector poses of both arms, and gripper states) at time $t$, and $\mathcal{A}_t$ is the corresponding bimanual action. We seek to learn a mapping:

$$\mathcal{F}_{\text{VLBiMan}} : (\mathcal{T}, \mathcal{D}, \mathcal{S}_{\text{new}}) \mapsto \{\widetilde{\mathcal{A}}_t^{\text{new}}\}_{t=1}^{T'}, \tag{1}$$

where $\mathcal{S}_{\text{new}}$ denotes a new scene containing instance-level object variations or rearrangements, and $\widetilde{\mathcal{A}}_t^{\text{new}}$ denotes the synthesized bimanual trajectory adapted to $\mathcal{S}_{\text{new}}$. To achieve this, we decompose the overall policy synthesis into reusable invariant modules and scene-adaptive variants. This requires solving three core challenges: (1) *Task-object semantic grounding*: aligning $\mathcal{T}$ with semantically-relevant objects $o_k$ in the scene via visual-language grounding, *i.e.*, learning a mapping $\mathcal{G} : \mathcal{T} \mapsto \{o_k\}_{k=1}^{K}$. (2) *Executable module decomposition*: partitioning $\mathcal{D}$ into temporally ordered motion primitives $\mathcal{M}_i$ with discrete boundaries $t_i$ such that each $\mathcal{M}_i$ is either task-invariant or requires adaptation. (3) *Trajectory composition with kinematic feasibility*: synthesizing a new trajectory $\widetilde{\mathcal{A}}_t^{\text{new}}$ by composing primitives under scene-aware geometric and kinematic constraints.

### 3.1  TASK-AWARE BIMANUAL DECOMPOSITION

To enable reusable and adaptable dual-arm skills, we begin by parsing the one-shot demonstration $\mathcal{D}$ into semantically meaningful and structurally reusable modules, which involves two sub-procedures: spatiotemporal segmentation and atomic skill extraction.

**Spatiotemporal Segmentation**. We record the one-shot demonstration using a third-person stereo RGB camera at 10 FPS, synchronously collecting dual-arm end-effector 6-DoF poses and gripper

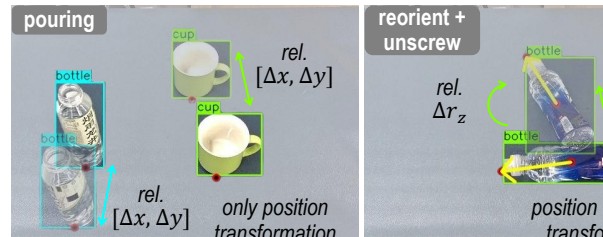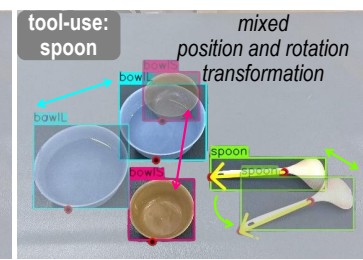

Figure 3: Illustrations of representative points for manipulated objects in three tasks: `pouring` (left), `reorient+unscrew` (middle) and `tool-use:spoon` (right). These points will be used to calculate the change in object position and orientation (not always required).

states. This forms a temporally aligned observation-action sequence: $\mathcal{D} = \{(\mathcal{O}_t, \mathcal{A}_t)\}_{t=1}^{T}$, where each action $\mathcal{A}_t \in \mathbb{R}^{14}$ consisting of 6-DoF for each arm with binary gripper states. We employ a keypose-driven segmentation scheme, which are inspired by those discrete motion prediction studies James et al. (2022); Shridhar et al. (2023); Ma et al. (2024); Ke et al. (2024). Initial segmentation can be scripted and automated via heuristics: trajectory waypoints are detected based on either changes in motion dynamics (*e.g.*, velocity discontinuities, acceleration spikes) or state switches (*e.g.*, gripper open/close transitions). Each candidate waypoint $\mathbf{w}_i$ divides the trajectory into time slots $\tau_i = [t_i, t_{i+1}]$. The inverse kinematics (IK) solver Chitta et al. (2012); Schulman et al. (2014) is used to validate the feasibility of trajectory segments $\mathcal{M}_i = \{\mathcal{A}_t\}_{t \in \tau_i}$.

Then, human-in-the-loop refinement ensures spatial continuity and execution robustness. Waypoints are inspected and manually adjusted in both temporal order and spatial distribution to guarantee smooth and robust control under the segmentation policy $\pi_{\texttt{seg}} : \mathcal{D} \to \{\mathcal{M}_i\}_{i=1}^{N}$.

**Atomic Skill Extraction**. To determine task-relevant modularity, we assign semantic labels to segments $\mathcal{M}_i$ by assessing object-robot couplings. For each $\mathcal{M}_i$, if no object is rigidly grasped, *i.e.*, object and end-effector are not in contact, the segment is classified as pre-contact adaptation dependent and potentially variable. Once the object is grasped and rigidly coupled with an end-effector (verified via gripper state and object pose consistency), subsequent motion is considered task-invariant, such as lifting or dual-arm alignment. Let $\texttt{bind}(o, r, t)$ be a binary indicator of whether object $o$ is physically attached to end-effector $r$ at time $t$. We define a skill $\mathcal{M}_i$ as invariant if:

$$\forall t \in \tau_i, \texttt{bind}(o_k, r, t) = 1, \text{and } \texttt{geometry}(o_k) \approx \texttt{geometry}(o_k^{\text{demo}}), \qquad (2)$$

where the $\approx$ denotes geometrically equivalent dimensions within a tolerance threshold $\epsilon_g$. Otherwise, we mark $\mathcal{M}_i$ as requiring adaptation. This yields a decomposition into:

$$\mathcal{D} \Rightarrow \{\mathcal{M}_i^{\text{inv}}\}_{i=1}^{N_{\text{inv}}} \cup \{\mathcal{M}_j^{\text{var}}\}_{j=1}^{N_{\text{var}}}. \qquad (3)$$

These atomic skill modules are stored for reuse and recomposition in novel scenes or tasks. Some illustrations on the pouring water task can be found in the left side of Fig. 2.

## 3.2 VISION-LANGUAGE ANCHORED ADAPTATION

Adaptation of variable modules $\mathcal{M}_j^{\text{var}}$ is anchored by semantic perception and geometric reasoning, structured into components: VLM-based scene understanding and VFM-based geometric feasibility.

**VLM-Based Scene Understanding**. We extract task-relevant prompts $p_k$ from the text description $\mathcal{T}$, mapping them to object categories. These are passed to the VLMs (*e.g.*, Florence-2 Xiao et al. (2024) and SAM2 Ravi et al. (2025)) to obtain high-quality 2D semantic masks $\mathbf{M}_k^{\text{2D}}$ from the current scene observation $\mathcal{O}^{\text{new}}$. Given the robustness of VLMs to lighting variations and distractors, we leverage their segmentation results to ground physical object identity without requiring explicit detection or prior 3D models.

**VFM-Based Geometric Feasibility**. To adapt grasping or alignment poses, we introduce a three-step process. (1) Firstly, we compute relative *position transformation* $\Delta\mathbf{T}$ between new object placement and reference demonstration via task-specific representative points (*e.g.* the 2D mask centroid or a task-specific contact point on the table-facing boundary). Examples of two kinds of representative points can be found in Fig. 3. Let $\mathbf{p}^{\text{demo}}, \mathbf{p}^{\text{new}}$ denote the representative 3D positions back-projected from 2D points via stereo and calibrated camera intrinsics. The relative position shift

is $\Delta\mathbf{x} = \mathbf{p}^{\text{new}} - \mathbf{p}^{\text{demo}}$. (2) To account for *orientation-sensitive* objects (such as pen, spoon and lying down bottle), we compute the principal axis from second-order image moments Chaumette (2004); Kotoulas & Andreadis (2007) of the 2D mask and derive relative rotation $\Delta\theta = \angle(\mathbf{v}^{\text{new}}, \mathbf{v}^{\text{demo}})$ via angular deviation. The final adapted grasp pose $\widetilde{\mathbf{T}}$ is obtained by applying $(\Delta\mathbf{x}, \Delta\theta)$ to the original grasp pose in robot coordinates via calibrated hand-eye transformation. (3) For *category-level variation*, we measure shape-induced feasibility change through height and width differences. For example, the $z$-extent of the object point cloud $\mathcal{P}_k^{\text{3D}}$ yields $\Delta h_k = \max_z(\mathcal{P}_k^{\text{3D}}) - \min_z(\mathcal{P}_k^{\text{3D}})$. This is used to adjust vertical placement motions or inter-arm distances for tools or containers.

Notably, we avoid applying 6-DoF pose estimation Lin et al. (2024); Wen et al. (2024a) or grasp pose detection Fang et al. (2020; 2023) methods in our adaptation, as they either depend on pre-defined CAD models or produce ambiguous non-semantic proposals, which are fragile and unfriendly.

### 3.3 Autonomous Trajectory Composition

After adaptation, we compose a new executable trajectory $\widetilde{\mathcal{D}}$ by aligning $\mathcal{M}_i^{\text{inv}}$ and $\widetilde{\mathcal{M}}_j^{\text{var}}$ according to the original temporal structure. However, this naive assembly may suffer from infeasibility due to reachability or collision. We therefore apply two refinements:

**Progressive IK Refinement**: For initial grasping motions $\widetilde{\mathcal{M}}_{\text{grasp}}$, we iteratively solve IK with interpolated splines approaching the target pose:

$$\mathbf{q}^{(n+1)} = \text{IK}(\mathbf{T}_g^{(n)}), \quad \mathbf{T}_g^{(n)} = \text{SplineInterp}(\mathbf{T}_{\text{start}}, \mathbf{T}_{\text{goal}}, n), \tag{4}$$

where $\mathbf{T}_{\text{start}}$ is the continuously updated initial pose, $\mathbf{T}_{\text{goal}}$ is the final goal that remains unchanged or is recalculated after being disturbed by external factors (such as human relocation or movement after being touched), and $n$ represents the interpolation density (which is set to 6 in our experiments). This refinement brings closed-loop correction under object displacement.

**Dynamic Collision Compensation**: To reduce early contact risks, we add proximal and vertical compensation terms $\delta_{\text{base}}$ and $\delta_z$ on the position item during grasp approach:

$$\tilde{\mathbf{x}}^{\text{goal}} = \mathbf{x}^{\text{goal}} + \delta_{\text{base}}\mathbf{u}_{\|} + \delta_z\mathbf{u}_z, \tag{5}$$

where $\mathbf{u}_{\|}$ and $\mathbf{u}_z$ respectively represent 3D Cartesian coordinates. After full trajectory synthesis, we perform one-time physical replay to observe unintended collisions and adjust motion plans accordingly. The adjusted plan remains reusable for repeated deployments of the identical object.

Thanks to modularity, VLBiMan supports cross-task module assembly and long-horizon tool-based task compositions by reusing $\mathcal{M}_i^{\text{inv}}$ across tasks. This enables not only generalization within a task, but also scalable extension to new task compositions, as illustrated in Fig. 5(b,c).

## 4 Experiments

We aim to answer following research questions: (1) How well does our framework automatically formulate and synthesize bimanual manipulation behaviors (Sec. 4.1)? (2) Can our method generalize to novel scenarios and achieve effective combination of skills (Sec. 4.2)? (3) How do individual components contribute to the effectiveness and robustness of our system (Sec. 4.3)? We validate VL-BiMan on a stationary dual-arm platform with two parallel grippers and a binocular camera (Fig. 4). Additional implementation details can be found in Supplementary Materials.

**Tasks and Setups**: We have designed up to 10 bimanual tasks (Fig. 5). In each task, at least two category-level objects with different geometric shapes are covered (Fig. 4), for comprehensively testing the performance in the face of novel placements and instances. These tasks involve diverse skill operations, complex multiple stages, and contact-rich tool-using, which can help to test the generalization. The external dynamic interference might be involved to check robustness.

**Baselines and Metric**: For each task setting, we conduct 25 trials, where objects are randomly located or replaced, and the success rate will be reported. For baselines, we compare to Robot-ABC Ju et al. (2024) based on keypoint affordance prediction with using AnyGrasp Fang et al. (2023) for initial grasping (After which, the remaining trajectory is obtained by trivial modules combination), as well as ReKep Huang et al. (2024b) based on VFMs (SAM Kirillov et al. (2023)

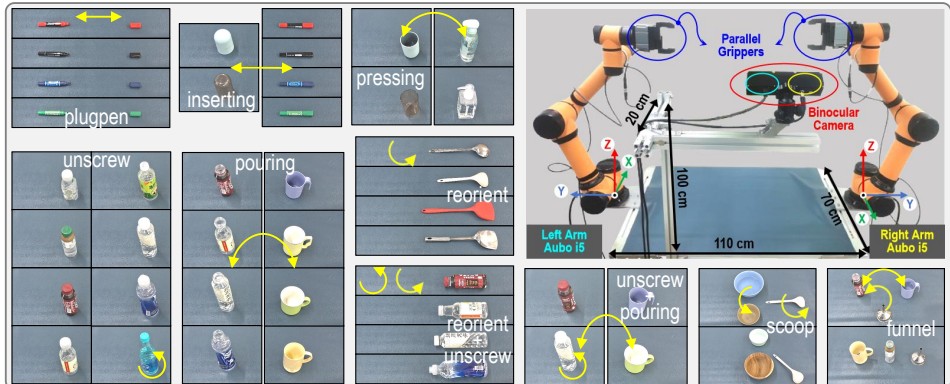

Figure 4: Manipulated object assets involved in each task, and the fixed-base dual-arm platform.

Table 1: Quantitative comparison results of success rates on **six primary bimanual tasks/skills**.

| Dynamic Inter-ference | Manipulation Method | *new placements + same objects* | | | | | | | *new placements + novel instances* | | | | | | |
|---|---|---|---|---|---|---|---|---|---|---|---|---|---|---|---|
| | | plugpen | inserting | unscrew | pouring | pressing | reorient | Average Success Rate | plugpen | inserting | unscrew | pouring | pressing | reorient | Average Success Rate |
| No | Mechanisms | 11/25 | 09/25 | 05/25 | 05/25 | 07/25 | 03/25 | 26.7% | 06/25 | 05/25 | 02/25 | 01/25 | 04/25 | 01/25 | 12.7% |
| | MAGIC | 16/25 | 15/25 | 10/25 | 10/25 | 09/25 | 07/25 | 44.7% | 11/25 | 10/25 | 05/25 | 05/25 | 06/25 | 04/25 | 27.3% |
| | Robot-ABC | 14/25 | 10/25 | 09/25 | 07/25 | 08/25 | 06/25 | 36.0% | 11/25 | 09/25 | 03/25 | 02/25 | 07/25 | 04/25 | 24.0% |
| | ReKep | 14/25 | 11/25 | 10/25 | 12/25 | 10/25 | 08/25 | 43.3% | 12/25 | 08/25 | 05/25 | 06/25 | 07/25 | 06/25 | 29.3% |
| | ReKep+ | 19/25 | 18/25 | 13/25 | 17/25 | 17/25 | 11/25 | 63.3% | 15/25 | 12/25 | 09/25 | 10/25 | 11/25 | 07/25 | 42.7% |
| | **VLBiMan** | 25/25 | 23/25 | 20/25 | 21/25 | 20/25 | 19/25 | **85.3%** | 24/25 | 21/25 | 18/25 | 17/25 | 20/25 | 17/25 | **78.0%** |
| Yes | Mechanisms | 05/25 | 05/25 | 03/25 | 02/25 | 04/25 | 01/25 | 13.3% | 03/25 | 01/25 | 00/25 | 00/25 | 02/25 | 00/25 | 4.0% |
| | MAGIC | 09/25 | 09/25 | 05/25 | 04/25 | 06/25 | 04/25 | 24.7% | 05/25 | 04/25 | 03/25 | 01/25 | 04/25 | 01/25 | 12.0% |
| | Robot-ABC | 07/25 | 06/25 | 04/25 | 03/25 | 05/25 | 02/25 | 18.0% | 05/25 | 03/25 | 00/25 | 00/25 | 03/25 | 00/25 | 7.3% |
| | ReKep | 10/25 | 06/25 | 06/25 | 04/25 | 05/25 | 03/25 | 22.7% | 09/25 | 04/25 | 03/25 | 02/25 | 04/25 | 02/25 | 15.3% |
| | ReKep+ | 12/25 | 10/25 | 09/25 | 08/25 | 09/25 | 09/25 | 38.0% | 10/25 | 08/25 | 05/25 | 04/25 | 06/25 | 05/25 | 25.3% |
| | **VLBiMan** | 19/25 | 16/25 | 19/25 | 18/25 | 17/25 | 15/25 | **69.3%** | 18/25 | 14/25 | 15/25 | 14/25 | 15/25 | 13/25 | **59.3%** |

and DINOv2 Oquab et al. (2024)) and GPT-4o Achiam et al. (2023). Besides, for a convincing comparison, an enhanced ReKep+ is introduced, where we inject an oracle-level initial grasp label to mitigate the impact of noisy perception. We also adapt two one-shot single-arm manipulation methods Mechanisms Mao et al. (2023) and MAGIC Liu et al. (2025b) for our dual-arm tasks.

## 4.1 EFFECTIVE AND ROBUST BIMANUAL MANIPULATION WITH VLBIMAN

Firstly, we compare to baselines on six basic dual-arm tasks as summarized on Tab. 1 left. In general, our VLBiMan shows promising capabilities and advantages in various complex situations, regardless of whether the interference is applied. For example, it can timely adjust the end-effector 6-DoF pose and achieve task-related precise grasping for unseen position and orientation of objects (including pens in and `inserting`, or spoons in `reorient`), which reflects the high success rate of the initial grasping stage. For actions that require fine-grained dual-arm coordination (such as aligning the pen tip and pen cap in `plugpen`, or aligning bottle mouth and cup mouth in `pouring`), it can always synthesize trustworthy trajectories to deal with these challenges. This ability benefits from decoupling and reusing invariant modules to the greatest extent. Strong baselines ReKep Huang et al. (2024b) and Robot-ABC Ju et al. (2024) do not have such a concept. For each new placement, they always need to re-plan the grasping and motion paths, which cannot fully explore and effectively utilize core components in a given demonstraion. The adapted baselines Mechanisms Mao et al. (2023) and MAGIC Liu et al. (2025b) originally designed for single-arm tasks also cannot handle these bimanual tasks well, revealing the non-trivial nature of dual-arm coordination.

## 4.2 GENERALIZATION ON NOVEL SCENARIOS AND SKILLS COMBINATION

To prove that VLBiMan has stronger generalization, such as being able to quickly transfer skills taught in a single time to new category-level objects, or further realize skills combination, complete complex multi-stage tool-use tasks, and transfer to other dual-arm robots. We conducted extensive experiments on six basic tasks (see Tab. 1 right) and four long-horizon tasks (see Tab. 2). The final results again show that VLBiMan has outstanding performance and significant advantages.

Table 2: Quantitative comparison results of success rates on **four long-horizon multi-stage tasks**.

| Dynamic Inter-ference | Manipulation Method | *new placements + same objects* | | | | | *new placements + novel instances* | | | | |
|---|---|---|---|---|---|---|---|---|---|---|---|
| | | reorient +unscrew | unscrew +pouring | tool-use scoop | tool-use funnel | Average Success Rate | reorient +unscrew | unscrew +pouring | tool-use scoop | tool-use funnel | Average Success Rate |
| No | Mechanisms | 05/25 | 04/25 | 02/25 | 01/25 | 12.0% | 01/25 | 02/25 | 00/25 | 00/25 | 3.0% |
| | MAGIC | 09/25 | 08/25 | 04/25 | 03/25 | 24.0% | 05/25 | 04/25 | 01/25 | 01/25 | 11.0% |
| | Robot-ABC | 06/25 | 06/25 | 03/25 | 03/25 | 18.0% | 04/25 | 02/25 | 00/25 | 01/25 | 7.0% |
| | ReKep | 07/25 | 08/25 | 05/25 | 03/25 | 23.0% | 05/25 | 04/25 | 01/25 | 00/25 | 10.0% |
| | ReKep+ | 11/25 | 10/25 | 07/25 | 06/25 | 34.0% | 07/25 | 06/25 | 04/25 | 02/25 | 19.0% |
| | **VLBiMan** | 15/25 | 15/25 | 12/25 | 10/25 | **52.0%** | 12/25 | 11/25 | 10/25 | 08/25 | **41.0%** |
| Yes | Mechanisms | 01/25 | 02/25 | 00/25 | 00/25 | 3.0% | 00/25 | 00/25 | 00/25 | 00/25 | 0.0% |
| | MAGIC | 04/25 | 03/25 | 04/25 | 01/25 | 12.0% | 02/25 | 02/25 | 01/25 | 00/25 | 5.0% |
| | Robot-ABC | 02/25 | 02/25 | 02/25 | 02/25 | 9.0% | 01/25 | 00/25 | 01/25 | 00/25 | 2.0% |
| | ReKep | 06/25 | 05/25 | 03/25 | 02/25 | 16.0% | 03/25 | 03/25 | 00/25 | 00/25 | 6.0% |
| | ReKep+ | 08/25 | 08/25 | 05/25 | 03/25 | 24.0% | 06/25 | 04/25 | 01/25 | 01/25 | 12.0% |
| | **VLBiMan** | 12/25 | 11/25 | 09/25 | 06/25 | **38.0%** | 08/25 | 09/25 | 05/25 | 02/25 | **24.0%** |

Figure 5: Visualization of ten tasks executed on real robots. They are designed to validate different aspects, including **(a)** six dual-arm primary skills, **(b)** combination of basic skills for two long-horizon tasks, and **(c)** exploration of two multi-stage tool-use tasks.

For example, in six basic tasks, it can correctly handle unseen objects according to the VLMs anchored adaptation, achieve stable combination of variable modules and readjusted invariant modules, and synthesize new executable trajectories. For long-horizon tasks, the first two are the sequential superposition of two basic tasks. The difficulty lies in that new intermediate grasping stages during the task execution are introduced (*e.g.*, in reorient+unscrew, the right arm needs to pick up and straighten the lying bottle first, and then the right arm takes the bottle to perform the dual-arm collaborative unscrewing of the bottle cap). These difficulties challenge the adaptability and multi-stage compatibility. The latter two tool-use tasks naturally contain additional common sense related to affordances, as well as multi-object contact-intensive actions, which introduce troubles including the organic connection of sub-modules and mutual interference of multiple objects. VLBiMan effectively alleviates these challenges with the help of powerful vision perception capabilities of VLMs and reasonable skill reuse design. While, baselines Mao et al. (2023); Liu et al. (2025b); Ju et al. (2024); Huang et al. (2024b) still perform poorly on these more complex dual-arm tasks. More importantly, we can still impose external interference on these long-horizon tasks, indicating VLBi-Man more practical and feasible. Fig. 5 shows visualization results. Besides, we migrated VLBiMan to a humanoid dual-arm robot to demonstrate its ability to generalize across different embodiment types. Qualitative results are shown in Fig. 6. Please refer to the Appendix for more details.

## 4.3 SYSTEM PERFORMANCE ABLATION AND ANALYSIS

Our modular solution has good process controllability and theoretical interpretability. We conducted the following two analyses on VLBiMan: ablation studies on module effectiveness and multi-factor statistics on system errors. First, we focused on four core designs (including VLMs type, initial grasp alignment, IK refinement, and collision avoidance), and checked the corresponding system performance. The results are shown in Tab. 3. It can be found that choosing the more advanced VLMs has obvious advantages, and our initial grasp adaptation scheme is more robust than the non-

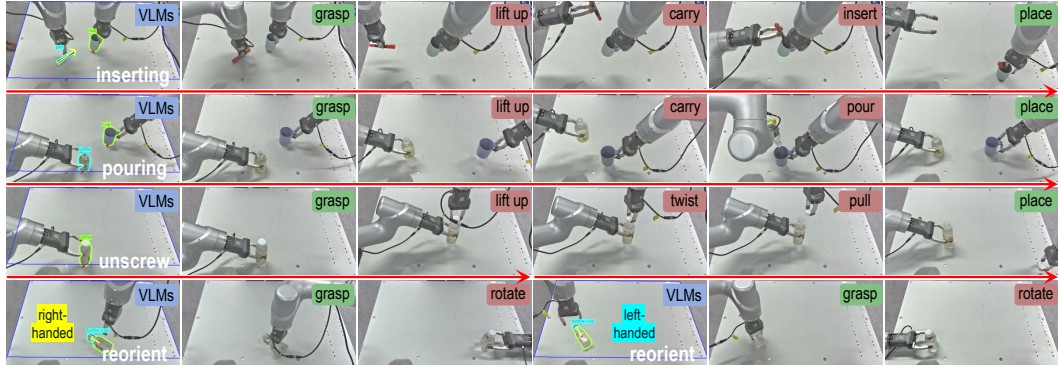

Figure 6: Visualization of four cross-embodiment transferred tasks executed on new humanoid arms.

semantic AnyGrasp Fang et al. (2023) (where we find the one closest to the demo grasp pose from many proposals for fair comparison). In addition, the kinematic optimization for trajectory synthesis is much better than the trivial module stacking, which is consistent with common sense.

Then, we conducted a statistical analysis of failed cases for results on Tab. 1 right (the interference part), and results are plotted in Fig. 7. The most prominent errors come from the initial grasp executing, even though its computing is relatively more reliable (with a lower error rate), which shows that performing task-related grasping in real-world is not easy, and there are a considerable proportion of singularity points or early collision problems. The second most error comes from the dual-arm coordination, which is the most core challenge of bimanual tasks. An optional mitigation solution is the closed-loop servo alignment. Finally, other items such as VLM-based perception and anchoring occupy a smaller proportion, indicating that it is at least reliable for our tasks, and the lower proportion of trajectory optimization indicates that the overall feasibility of our solution is well. Through these exhaustive analyses, we can understand the advantages and defects of VLBiMan.

Table 3: Ablation studies of VLBiMan. All trials were completed on six basic tasks, under the *new placements + novel instances* evaluation, with interference.

| VLMs type | initial grasp alignment | IK refinement | collision avoidance | Avg. SR |
|---|---|---|---|---|
| SAM+DINOv2 | ours | ✓ | ✓ | 35.8% |
| ours | AnyGrasp | ✓ | ✓ | 31.7% |
| ours | ours | ✗ | ✓ | 29.2% |
| ours | ours | ✓ | ✗ | 34.2% |
| ours | ours | ✓ | ✓ | 59.2% |

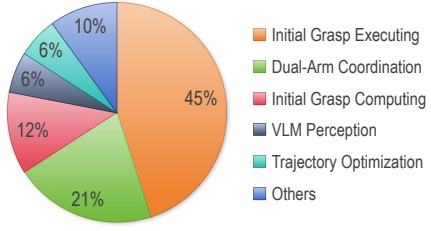

Figure 7: Error breakdown of VLBiMan.

## 5 CONCLUSION AND LIMITATION

In this work, we present VLBiMan, a novel framework that enables generalizable bimanual manipulation from a single human demonstration, guided by a natural language task description. Through a task-aware decomposition strategy, vision-language grounded scene understanding, and geometric adaptation anchored by visual representations, our approach efficiently composes executable bimanual trajectories under diverse scene variations. Without reliance on object-specific priors or pose annotations, VLBiMan achieves robust generalization across unseen object instances and another dual-arm robots. Extensive experiments demonstrate its effectiveness across a wide range of real-world bimanual tasks, including tool use, and long-horizon compositions.

**Limitations**: Despite promising results, VLBiMan still faces several limitations. First, it is restricted to rigid objects and does not handle deformable items such as cloth or rope, which require different representations and control. Second, it lacks runtime anomaly detection and recovery mechanisms, making it sensitive to execution errors like slippage or occlusion. Third, the capability of our approach is inherently bounded by the hardware: the fixed-base dual-arm platform limits the reachable workspace and lacks force or tactile sensing. Future work could explore extending the system to a mobile base to enhance spatial flexibility, and equipping end-effectors with force or tactile sensors to enable fine manipulation of delicate or force-sensitive objects.

ACKNOWLEDGMENTS

This work was supported by the Guangdong Provincial Key Field R&D Program (Project No. 20240104), and was also funded by the Shenzhen Science and Technology Major Project (No. 202402002 and ZDCT20250901113000001).

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

APPENDIX

This supplementary part provides detailed clarifications and additional insights to support the main paper. In Sec. A (**Discussion on Bimanual Manipulation Tasks**), we present the motivation behind the design of ten bimanual manipulation tasks, including an overview of the dual-arm robotic platform, a task-by-task breakdown, and the process of collecting one-shot demonstrations for each task. In Sec. B (**More Implementation Details of VLBiMan**), we elaborate on the implementation details of the proposed VLBiMan framework, such as the object principal axis extraction algorithm based on image moments, and procedure for robust dual-arm execution under external dynamic disturbances. In Sec. C (**More Exploration on VLBiMan Advantages and Limitations**), we explore further strengths of VLBiMan, including its robustness to lighting variations and its modular structure, which allows for synchronous dual-arm sub-skills to improve manipulation efficiency. We also provide additional experimental results and analyses, such as evaluating the impact of varying levels of external interference on task success rates, revealing the ease with which the system can be transferred across embodiments, as well as discussing some interesting empirical findings. This section also includes additional analyses introduced in four new subsections: (1) a detailed discussion of the human-in-the-loop refinement process used during primitive segmentation, clarifying its role and negligible burden; (2) an investigation into the robustness of using simple object representing points—such as mask centroids or front-edge contacts—for cross-object generalization; (3) evaluations of VLBiMan under cluttered scenes to assess stability in more realistic environments; and (4) ablation studies on pre-grasp interpolation density, highlighting its effect on collision avoidance and resilience to external disturbances. Finally, Sec. D is the statement on the use of LLMs.

## A    DISCUSSION ON BIMANUAL MANIPULATION TASKS

### A.1    FIXED-BASE DUAL-ARM PLATFORM

Our manipulation platform consists of a rectangular tabletop approximately 110 cm in length and 70 cm in width, equipped with two fixed-base robotic arms, parallel grippers, and a binocular vision system (see Fig. 4 in the main paper for layout). The dual arms are mounted on opposite short edges of the table. This is an opposite-side configuration, which differs from the more common same-side or humanoid-style arrangements. This design significantly reduces workspace overlap between the arms, thereby expanding their combined reachable workspace. The trade-off, however, is a reduced resemblance to human-like coordination patterns. Each arm is mounted at the center of the table short edge, with its base extended slightly beyond the tabletop to save space.

The manipulators are Aubo-i5 collaborative robots[1] (880mm reach) with six degrees of freedom and a maximum reach of approximately 880 mm. Note that these arms do not feature built-in force control at the joints. Each arm is equipped with a DH-Robotics parallel gripper[2], offering a maximum width of 80 mm and an effective finger length of about 50 mm (total length approximately 160 mm, used to compensate for tool flange length). While the gripper can be controlled at arbitrary open ratios, we restrict it to two discrete states (open and closed) across all experiments. For visual perception, we employ a binocular Kingfisher R-6000 stereo camera, capturing RGB images at 960×540 resolution and supporting 3D scene reconstruction via calibrated stereo intrinsics. This setup functions similarly to standard RGB-D cameras, but offers improved reconstruction quality and greater flexibility through algorithm-level customization. The camera is mounted in a third-person perspective, positioned approximately 20 cm horizontally and 100 cm vertically from one of the long edges of the table, enabling full coverage of the workspace. Consequently, we do not employ eye-in-hand cameras at the robot end-effectors. To further demonstrate the convenient transferability of VLBiMan, as shown in Fig. 8, we have prepared another dual-arm robotic platform configured in a popular humanoid style. This new platform consists of two Rokae xMate CR7[3] 6-DoF collaborative arms (reach: 988 mm), each equipped with a parallel gripper (Jodell Robotics RG75-300[4], max opening: 75 mm). A binocular camera Kingfisher R-6000 is mounted centrally at the head position. We will present how to utilize this dual-arm platform in Sec. C.4.

---

[1]https://www.aubo-cobot.com/public/i5product3
[2]https://en.dh-robotics.com/product/pg
[3]https://www.rokae.com/en/product/show/545/xMateCR.html
[4]https://www.jodell-robotics.com/product-detail?id=5

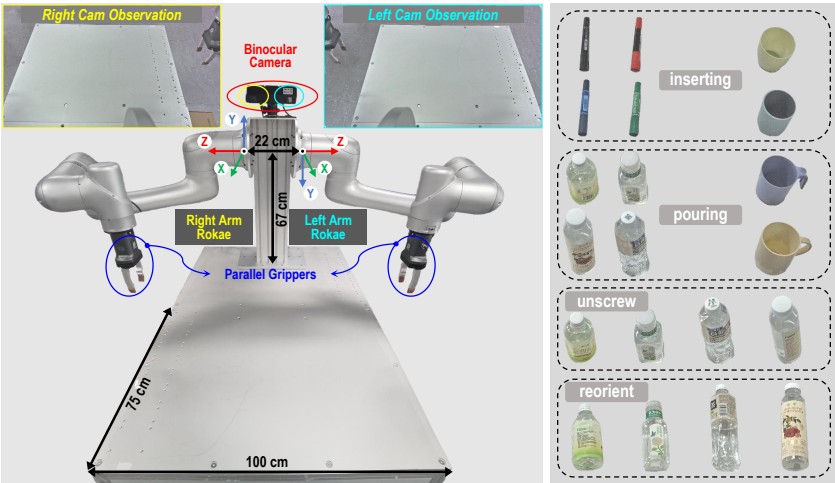

Figure 8: The another dual-arm manipulator platform (left) and corresponding manipulated object assets (right) used for the cross-embodiment evaluation.

## A.2 INTRODUCTION TO BIMANUAL TASKS

To comprehensively evaluate the dual-arm manipulation capabilities of our system, we first design a suite of six foundational bimanual tasks: `plugpen`, `inserting`, `unscrew`, `pouring`, `pressing` and `reorient`. Each task involves manipulating objects drawn from at least two category-level instances (see Fig. 4 in the main paper), enabling systematic assessment the generalization performance of VLBiMan under object instance variations. These tasks encompass a broad range of atomic manipulation skills, such as single-arm actions like grasping, placing, inserting, transporting, pressing, and precise reorientation, as well as coordinated dual-arm behaviors including fix-and-unscrew, fix-and-skew-insert, bilateral alignment, and handover. Together, they ensure sufficient complexity and coverage of real-world manipulation demands.

- `plugpen`: *Plug the marker body into its cap*. The task begins with a separated pen body and cap placed on the table. The left and right arms grasp the pen body and cap respectively, lift them, align the pen tip with the cap opening, and perform a high-precision plug-in motion to close the pen. The right gripper then releases, and the left arm places the assembled pen on the table. This task demands accurate segmentation of small objects, orientation-aware grasping, and near-zero-tolerance insertion. To avoid issues due to the lack of eye-in-hand cameras, configurations where the pen tip or cap opening faces downward are excluded.

- `inserting`: *Insert a closed marker pen into an inverted cup*. The setup includes an upside-down, handleless cup and a fully assembled marker pen. The left and right arms grasp the cup and pen respectively, lift them, and the left arm rotates the cup to face upward. Simultaneously, the right arm reorients the pen vertically for insertion. After aligning the two objects, the right arm inserts the pen, releases it, and the left arm places the cup back. The task requires precise rotation for object reorientation, orientation-aware grasping, and moderate-tolerance insertion.

- `unscrew`: *Open a bottle by twisting the cap counterclockwise*. A sealed plastic bottle containing water stands upright on the table. The left arm grasps and lifts the bottle, holding it steady in mid-air. The right arm approaches the cap vertically from above, grasps it, and performs multiple controlled counterclockwise rotations to unscrew it. The cap is then placed on the table, and the bottle is set down. This task involves extremely tight grasping tolerance (for the cap), precise rotational control, and potentially force-sensitive unscrewing (though our gripper lacks force sensing, which may increase the failure risk).

- `pouring`: *Pour water from a bottle into a mug*. A water-filled plastic bottle without a cap and an empty mug with a handle are placed on the table. The left and right arms grasp the bottle and mug respectively, lift them, and coordinate to align the bottle and mug openings. The left arm rotates approximately $90°$ to pour water, then restores the bottle to an upright position. The right arm retracts the filled mug, and both objects are returned to the table. This task requires moderate-tolerance alignment, precise angular control for pouring, and careful handling of the deformable bottle body (deformation may affect the bottle's geometry and induce spill errors).

- `pressing`: *Press a pump bottle and catch the water in a cup*. A shampoo bottle with a pressable nozzle and a handleless upward-facing cup are provided. The left arm grasps the cup and lifts it to a tilted receiving position near the nozzle. The right arm approaches vertically and presses the nozzle to dispense a small amount of water. Both arms then place the objects back. Key challenges include precise nozzle approach, accurate cup positioning for water collection, and robust force application (although without force feedback, we rely on a fixed press depth that balances functionality and bottle safety).

- `reorient`: *Flip a spoon or shovel so that its concave side faces upward*. The object starts in an arbitrary pose with the concave side down. The right arm vertically grasps its center and lifts it, then repositions and reorients it into a graspable pose for the left arm. The left arm then grasps the handle, the right arm releases, and the left arm completes the flipping motion to place the spoon upright on the table. This task demands precise reorientation, spatially and temporally coordinated handover, and potentially strong grasping (the left arm's handle grasp is relatively unstable and may result in object slippage during motion).

In addition to the six base tasks, we introduce four more challenging long-horizon tasks to evaluate VLBiMan's capacity for skill composition and multi-stage adaptive control. The tasks include `reorient+unscrew`, `unscrew+pouring`, `tool-use spoon`, and `tool-use funnel`. The first two are about concatenations of previously defined skills, while the latter two require tool-use behaviors that test the system's ability to generalize across distinct affordances.

- `reorient+unscrew`: *Straighten a fallen bottle and unscrew its cap*. A sealed water bottle lies horizontally on the table. The right arm vertically grasps and reorients the bottle upright, ensuring the cap faces upward. The system then proceeds with the unscrewing routine. The new challenge lies in accurate estimation of the lying down bottle's orientation, especially the cap direction.

- `unscrew+pouring`: *Open the bottle cap and pour water into a mug*. A sealed water-filled bottle and an empty mug are provided. The system first performs the full unscrewing sequence, followed by the water-pouring procedure. While the skills themselves are known, the compound task increases complexity through potential error accumulation across stages.

- `tool-use spoon`: *Use a spoon to transfer water from a larger bowl to a smaller one*. Three objects are involved: an upside-down spoon, a large bowl filled with water, and a smaller empty bowl. The system first performs reorientation on the spoon, then uses it to scoop water from the large bowl, transport it, and pour it into the smaller bowl before returning the spoon. The task requires multiple precise reorientation and motion sequences, handling mutual visual distractions among objects, and reliably distinguishing the bowls of different sizes.

- `tool-use funnel`: *Use a funnel to pour water from a mug into an empty bottle*. The setup includes an upside-down metal funnel, an empty plastic bottle without a cap, and a water-filled mug. The system reorients the funnel, inserts its narrow end into the bottle, then the left arm relocates the bottle near the right arm. The right arm lifts the mug and pours water into the bottle throught the funnel. This task tests multi-object coordination, spatial reasoning under occlusion, and moderate-tolerance insertion.

### A.3 One-shot Human Demonstration

We collect one-shot seed demonstrations for each task using kinesthetic teaching, wherein the operator manually guides the dual-arm robot to designated waypoints. Specifically, the full trajectory is decomposed into sparse keypoints by physically dragging the robotic arms to target poses. At each pause, we record the 6-DoF end-effector poses of both arms (relative to their respective robot coordinate frames) using a teach pendant, along with the intended gripper open/close states. Subsequently, with objects placed at approximately fixed initial positions, the robot autonomously replays the demonstration by executing the recorded sequence of waypoints. We first move the arms via inverse kinematics (handled by the control API), followed by gripper actions. Throughout this replay, we record synchronized observations from the binocular camera and the corresponding end-effector states at 10Hz. After collecting the demonstration, we decompose it using the task-aware strategy described in the main paper, enabling downstream skill reuse. Notably, for the composed tasks `reorient+unscrew` and `unscrew+pouring`, which are essentially combinations of existing base skills, we do not provide additional one-shot demonstrations, as their behavior can be sufficiently inferred from the constituent components.

---

**Algorithm 1** Overall Procedure of Object Orientation Estimation from 2D Mask.

---

**Require:** Binary object mask $\mathbf{M} \in \{0,1\}^{H \times W}$
**Ensure:** Orientation angle $\theta \in [0, 360)$ (degrees)
1: Extract object contour $C = \{(x_i, y_i)\}_{i=1}^{N}$ from $\mathbf{M}$
2: Compute centroid $(\bar{x}, \bar{y})$ using image moments:

$$\bar{x} = \frac{1}{N} \sum_{i=1}^{N} x_i, \quad \bar{y} = \frac{1}{N} \sum_{i=1}^{N} y_i \tag{6}$$

3: Calculate second-order central moments:

$$\mu_{20} = \frac{1}{N} \sum (x_i - \bar{x})^2, \quad \mu_{02} = \frac{1}{N} \sum (y_i - \bar{y})^2, \quad \mu_{11} = \frac{1}{N} \sum (x_i - \bar{x})(y_i - \bar{y}) \tag{7}$$

4: Construct covariance matrix: $\Sigma = \begin{bmatrix} \mu_{20} & \mu_{11} \\ \mu_{11} & \mu_{02} \end{bmatrix}$
5: Compute eigenvalues $\lambda_1 > \lambda_2$ and eigenvectors $\mathbf{v}_1, \mathbf{v}_2$ of $\Sigma$
6: Obtain principal axis direction $\mathbf{a} = (a_x, a_y) = \mathbf{v}_1$
7: Project contour points onto principal axis: $p_i = (x_i - \bar{x})a_x + (y_i - \bar{y})a_y \quad \forall i \in [1, N]$
8: Identify endpoints: $e_{max} = \arg\max_i p_i, \quad e_{min} = \arg\min_i p_i$
9: Calculate perpendicular width $w_j$ within radius $r$ around each endpoint $e_j$
10: Determine front endpoint: $e_{front} \leftarrow (w_{max} < w_{min})? e_{max} : e_{min}$
11: Adjust axis direction: $\mathbf{a} \leftarrow (\mathbf{a} \cdot (e_{front} - (\bar{x}, \bar{y})) < 0)? -\mathbf{a} : \mathbf{a}$
12: Compute final orientation angle: $\theta = (\arctan 2(a_y, a_x) \times \frac{180}{\pi}) \bmod 360$
13: **return** $\theta$

---

Table 4: Statistical details regarding the ten bimanual manipulation tasks defined in this study. They contain the names of the target objects involved (including their placement states), the representation points used to indicate the positions of the target objects (where MC represents the object's 2D mask centroid, and CP represents the contact point between the object bottom and the table top. Please refer to Fig. 3 in the main text for visualization), and whether the object's orientation needs to be estimated during the manipulation process.

| Task Name | plugpen | | inserting | | unscrew | pouring | | pressing | | reorient | reorient +unscrew | unscrew +pouring | | tool-use scoop | | | tool-use funnel | | |
|---|---|---|---|---|---|---|---|---|---|---|---|---|---|---|---|---|---|---|---|
| Object Name | marker body | marker cap | marker pen | cup | standing bottle | standing bottle | mug | cup | pump bottle | spoon or shovel | lying bottle | standing bottle | mug | spoon | big bowl | small bowl | funnel | standing bottle | mug |
| Representive Point Type | MC | MC | MC | CP | CP | CP | CP | CP | CP | MC | MC | CP | CP | MC | CP | CP | CP | CP | CP |
| Orientation Estimation? | ✓ | ✓ | ✓ | ✗ | ✗ | ✗ | ✗ | ✗ | ✗ | ✓ | ✓ | ✗ | ✗ | ✓ | ✗ | ✗ | ✓ | ✗ | ✗ |

## B  MORE IMPLEMENTATION DETAILS OF VLBiMAN

### B.1  IMAGE MOMENTS BASED ORIENTATION ESTIMATION

In the Vision-Language Anchored Adaptation pipeline of VLBiMan, our method requires extracting the principal axis and determining the orientation of direction-sensitive objects. This includes the marker pen in the `plugpen` and `inserting` tasks, the spoon in the `reorient` and `tool-use spoon` tasks, as well as the horizontally placed bottle in the `reorient+unscrew` task. As shown in Algorithm 1, we adopt an object principal axis extraction algorithm based on image moments theory Chaumette (2004); Kotoulas & Andreadis (2007). Since this algorithm relies primarily on the 2D segmentation mask of object and does not require any deep networks, its computational overhead is minimal and can be considered negligible in practice.

In general, the proposed algorithm estimates the orientation angle of an object from its 2D binary mask through a hierarchical analysis of geometric properties. First, the object's contour is extracted, and its centroid is computed using image moments. A covariance matrix derived from second-order central moments is then diagonalized to identify the principal axis direction via eigen decomposition. To resolve directional ambiguity inherent to eigenvectors, contour points are projected onto the

principal axis to locate two extreme endpoints. The front endpoint is determined by comparing local perpendicular widths around these endpoints, leveraging the observation that structural asymmetry often manifests as width variation. Finally, the principal axis direction is reoriented to align with the front endpoint, and the orientation angle is calculated as the arctangent of the adjusted axis vector, ensuring a continuous $0°$–$360°$ representation. This approach robustly handles directional ambiguity while maintaining computational efficiency through moment-based feature extraction. For more details on which objects in which tasks require the orientation estimation algorithm, please refer to Tab. 4 (see the last row).

**Why Not Use Off-the-Shelf 6D Pose Estimation?** While one may consider leveraging off-the-shelf 6D pose estimators Lin et al. (2024); Wen et al. (2024a) for object orientation extraction, we found that such solutions are unnecessary, unstable across objects, and incompatible with VLBi-Man's cross-object generalization objective. Classical and learning-based 6D pose estimators generally require either (1) object-specific CAD models, (2) textured templates, or (3) category-level canonicalization priors. These assumptions are difficult to satisfy in our setup, where VLBiMan must generalize to unseen and shape-diverse everyday objects without additional training or model registration. Moreover, 6D estimators often degrade when objects lack distinctive geometry or texture—precisely the case for many household items used in our tasks (e.g., plain spoons, cylindrical pens). In contrast, our moment-based orientation estimation (Algorithm 1) only depends on the 2D segmentation mask produced by a VLM-powered perception module, eliminating the need for any object-specific shape information. This makes the approach far more robust to appearance variations and naturally compatible with VLBiMan's object-centric anchoring framework. Additionally, we observed in our experiments that 6D pose estimators frequently output unstable yaw angles under partial occlusion or when only a single RGB camera view is available, while the moment-based method remains consistent, lightweight, and easy to deploy in real-world bimanual settings.

Finally, this simplified 2D-mask–driven orientation strategy is fully aligned with VLBiMan's design principle—to avoid heavy perception modules that compromise generalization—and it has proven sufficient for all direction-sensitive tasks, including `plugpen`, `inserting`, `reorient`, `tool-use spoon`, and `reorient+unscrew`. Hence, the choice to avoid 6D pose estimation is both practical and necessary: our goal is not to recover a full metric pose, but to obtain a stable, VLM-compatible orientation anchor that enables one-shot bimanual manipulation without retraining or object modeling.

## B.2 DYNAMIC INTERFERENCE ROBUST VLBIMAN

Thanks to the modular design of our VLBiMan system, we enable dynamic interference to be applied to an object before it is physically grasped by the robot arms (that is before the object formally becomes part of a robot-object composite system). Such interference may include randomly perturbing the position or orientation of the object multiple times, without any predefined limit, until the object is successfully captured. This capability introduces significant challenges for maintaining robustness during execution, requiring a carefully structured control process to ensure system reliability under disturbance. To address this, we summarize a dynamic closed-loop control pipeline tailored for interference robustness pre-grasping below.

Specifically, for each object to be manipulated, VLBiMan first performs continuous 2D instance segmentation and tracks the object across frames using a lightweight vision pipeline. Let $\mathbf{M}_t$ denote the segmentation mask at time step $t$, and let the corresponding object pose estimation function be $\mathcal{F}_{\texttt{pose}} \rightarrow (\mathbf{p}_t, \theta_t)$, where $\mathbf{p}_t$ is the 2D position and $\theta_t$ is the principal axis orientation obtained via image moments (refer Algorithm 1). This estimation is continuously updated and serves as input to the grasp planning module. A grasping attempt is initiated only when the variance of $\{\mathbf{p}_{t-k}, \cdots, \mathbf{p}_t\}$ and $\{\theta_{t-k}, \cdots, \theta_t\}$ over a short sliding window falls below a pre-defined threshold $\epsilon$ (e.g., absolute moving distance less than 10mm), indicating that the object has stabilized. This implicitly filters out moments of dynamic perturbation. Once the object is deemed stable, the robot executes the corresponding grasp action $\mathcal{G}(\mathbf{p}_t, \theta_t)$, where $\mathcal{G}(\cdot)$ denotes a grasp generation function conditioned on both position and orientation. If the grasp fails (e.g., the object slips or moves significantly post-action), the system returns to the observation loop and restarts the stabilization-checking process. This mechanism ensures that the object's interaction policy is dynamically robust, without requiring hard-coded assumptions on when or how disturbances may occur.

Table 5: Quantitative comparison results of success rates on **six primary bimanual skills/tasks**.

| Dynamic Interference + **Uneven Lighting** | Manipulation Method | *new placements + same objects* | | | | | | | *new placements + novel instances* | | | | | | |
|---|---|---|---|---|---|---|---|---|---|---|---|---|---|---|---|
| | | plugpen | inserting | unscrew | pouring | pressing | reorient | Average Success Rate | plugpen | inserting | unscrew | pouring | pressing | reorient | Average Success Rate |
| Yes | Mechanisms | 01/20 | 01/20 | 00/20 | 01/20 | 01/20 | 00/20 | 3.3% | 00/20 | 00/20 | 00/20 | 00/20 | 00/20 | 00/20 | 0.0% |
| | MAGIC | 03/20 | 04/20 | 02/20 | 02/20 | 02/20 | 01/20 | 11.7% | 01/20 | 02/20 | 00/20 | 01/20 | 01/20 | 00/20 | 4.2% |
| | Robot-ABC | 02/20 | 02/20 | 01/20 | 01/20 | 01/20 | 01/20 | 6.7% | 00/20 | 00/20 | 00/20 | 00/20 | 00/20 | 00/20 | 0.0% |
| | ReKep | 05/20 | 03/20 | 02/20 | 01/20 | 02/20 | 02/20 | 12.5% | 03/20 | 01/20 | 01/20 | 01/20 | 01/20 | 00/20 | 5.8% |
| | ReKep+ | 08/20 | 06/20 | 04/20 | 03/20 | 04/20 | 05/20 | 25.0% | 05/20 | 02/20 | 03/20 | 02/20 | 03/20 | 01/20 | 13.3% |
| | **VLBiMan** | 14/20 | 13/20 | 14/20 | 15/20 | 14/20 | 11/20 | 67.5% | 13/20 | 11/20 | 11/20 | 10/20 | 12/20 | 10/20 | 55.8% |

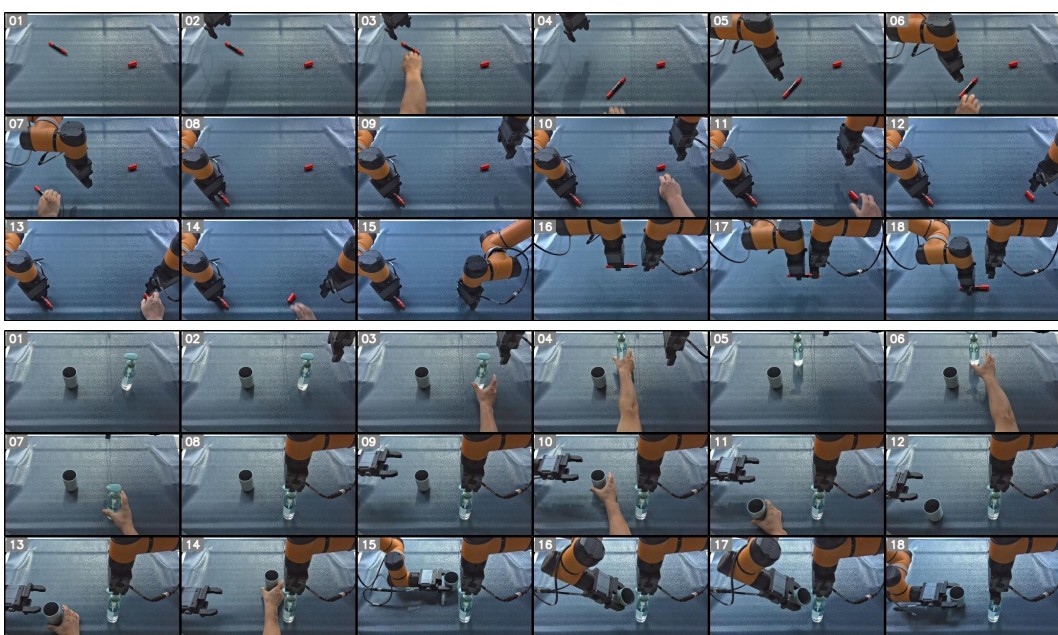

Figure 9: Examples of plugpen (*top*) and pressing (*bottom*) show that under the uneven lighting, the system is subjected to consecutive external interferences, and tasks can still be completed.

# C    MORE EXPLORATION ON VLBiMan ADVANTAGES AND LIMITATIONS

## C.1    GOOD ROBUSTNESS TO LIGHTING CHANGES

In addition to the generalization capabilities of VLBiMan with respect to spatial object positions and category-level instance variations, as demonstrated in the main text, we further explore another crucial advantage—its robustness to lighting changes, which also constitutes an important aspect of generalizable bimanual manipulation. Specifically, we investigate the impact of uneven illumination on task success rates. For this purpose, we evaluate six basic bimanual tasks under a setting where dynamic object perturbations are applied during the initial grasping phase, while also introducing uneven lighting conditions. These lighting conditions cause non-uniform brightness across the scene and cast shadows on the manipulated objects, posing new challenges to both the visual perception module and the grasp pose alignment procedure for our VLBiMan.

Thanks to the strong generalization ability of the VLMs Xiao et al. (2024) and VFMs Ravi et al. (2025) employed in our system, the detection and segmentation of target objects remain highly reliable even under such adverse lighting. Furthermore, our method for estimating object position and orientation relies solely on binary masks, which are inherently invariant to lighting variations. Quantitative and qualitative results under this setting are summarized in Tab. 5 and Fig. 9, respectively. As shown, *the effect of uneven illumination on VLBiMan's task success rate is minimal* (70.0% → 67.5% for ID testing, and 59.2% → 55.8% for OOD testing).

In contrast, two baselines Mechanisms Mao et al. (2023) and MAGIC Liu et al. (2025b) have had obvious negative effects (13.3% → 3.3% and 3.3% → 0.0% for Mechanisms, and 24.2% → 11.7% and 11.7% → 4.2% for MAGIC). For another two stronger baseline methods (Robot-ABC Ju et al.

(2024) and ReKep Huang et al. (2024b)) also exhibit substantial performance degradation (18.3% → 6.7% and 6.7% → 0.0% for Robot-ABC, 23.3% → 12.5% and 14.2% → 5.8% for ReKep, and 37.5% → 25.0% and 25.0% → 13.3% for ReKep+). This is not unexpected, as both baselines rely on inferior vision pipelines that are sensitive to lighting. For instance, AnyGrasp Fang et al. (2023), which Robot-ABC depends on for grasping, was never trained on point clouds data containing uneven illumination, and ReKep employs a fragile keypoints tracking mechanism that becomes prone to false positives and missed detections under such lighting variations. Additional dynamic execution records are available in our **supplementary videos**.

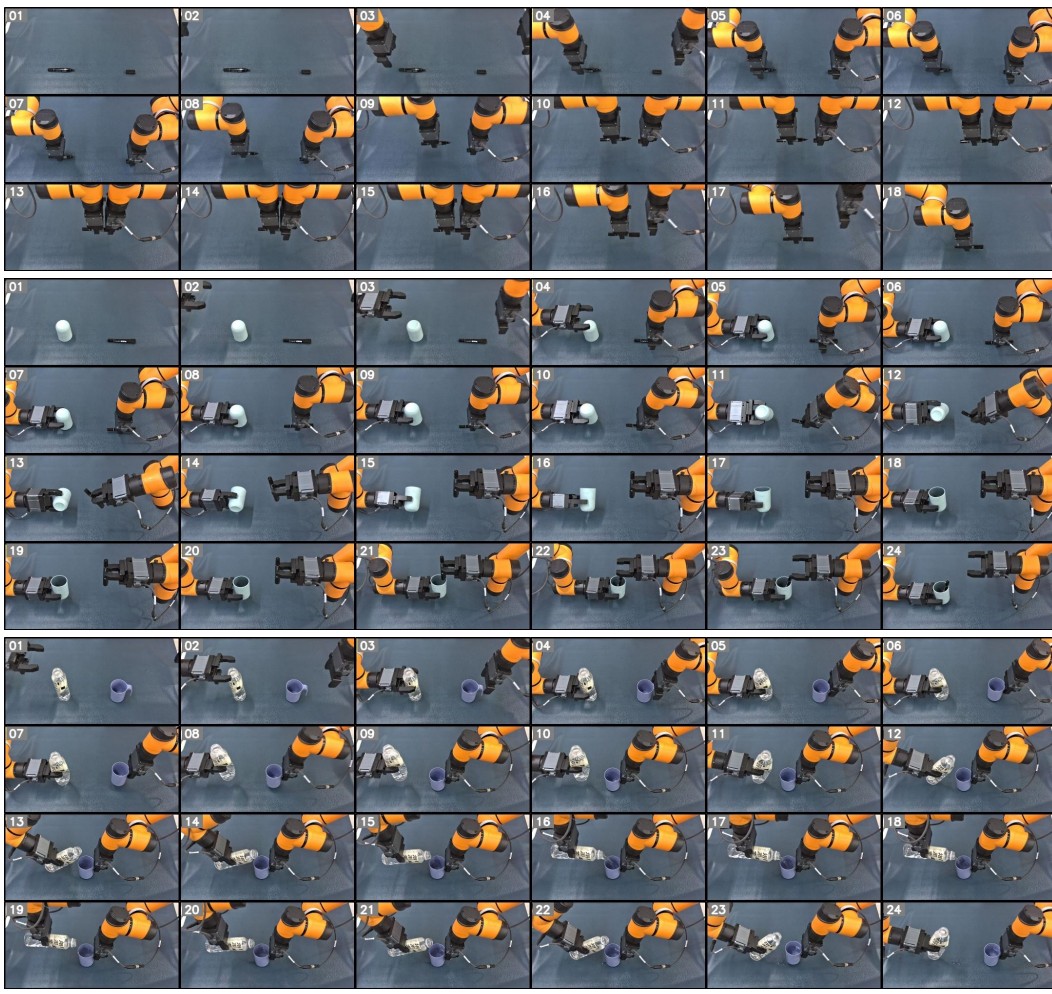

Figure 10: Examples of synchronized dual-arm movement. Segments from top to bottom are tasks `plugpen`, `inserting`, and `pouring`, which have relatively high dual-arm synchronizability.

## C.2 EFFICIENT SYNCHRONOUS DUAL-ARM MOVEMENT

Another notable advantage of the VLBiMan system lies in its ability to support more human-like dual-arm behaviors, specifically the hybrid execution of asynchronous and synchronous arm movements. This capability not only contributes to the overall manipulation efficiency of each task but also serves as an essential factor for achieving generalizable bimanual manipulation. For instance, certain tasks, such as lifting large balls Grotz et al. (2024); Liu et al. (2024a) or bimanual occluded grasping Yamada et al. (2025) (which we have not yet explored in this study), require strictly synchronized dual-arm motions.

In our system, after decomposing the given one-shot demonstration, we obtain temporally indexed motion sequences for both arms. These sequences are further processed with collision-avoidance strategies under a global trajectory perspective, enabling the possibility of triggering specific motion segments concurrently. For example, in the `plugpen` task, the left and right arms can move simul-

taneously to align and close the pen body and cap; similarly, in the `pouring` task, both arms can coordinate to bring the bottle and cup closer and align them for fluid transfer. Such synchronized execution clearly reduces overall task duration. However, this does not imply that the task execution time is halved, as certain motion segments inherently require strictly asynchronous behavior. For example, in the `unscrew` task, the left arm must serve as a stabilizer to hold the bottle stationary while the right arm unscrews the cap, making simultaneous execution infeasible.

To evaluate this advantage quantitatively, we conducted comparative experiments on all ten bimanual tasks, testing strictly asynchronous execution versus a strategy that maximally leverages synchronous execution wherever feasible. We observed time savings of varying magnitudes across all ten tasks, yielding *an average improvement in execution efficiency of approximately 22%*. Fig. 10 visualizes the synchronous motion segments for some tasks, and additional dynamic comparison footage can be found in our **supplementary videos**.

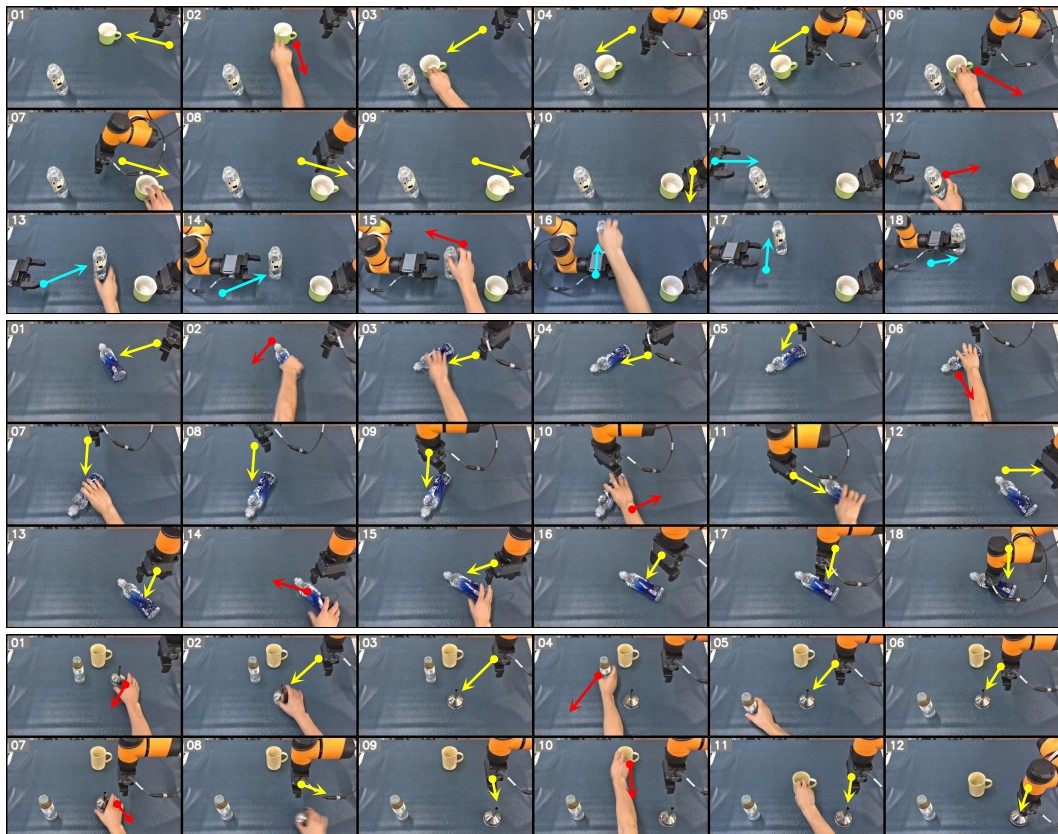

Figure 11: Example of dynamic interferences during task execution. From top to bottom, they are segments of the dynamic closed-loop grasping phase of tasks `pouring`, `reorient+unscrew`, and `tool-use funnel`, where each object is manually disturbed from one to three times. The red arrow indicates the direction of the manually moved object (interfering). The cyan arrow and yellow arrow indicate the movement direction of the left and right robotic arms (chasing) respectively.

### C.3 INTERFERENCE FREQUENCY AND SUCCESS RATE

We have extensively examined VLBiMan's robustness to external disturbances in both the main paper and this supplementary material, which is an essential capability for achieving highly generalizable bimanual manipulation. As discussed in works like AnyGrasp Fang et al. (2023), dynamic grasping presents substantial practical value while remaining a formidable challenge, even for systems already equipped with strong static 6-DoF grasp pose prediction and execution capabilities. Moreover, dynamic interference robustness also provides the embodied agent with a foundation for error recovery and correction mechanisms during execution, whether through end-to-end learning pipelines Black et al. (2024); Liu et al. (2025a); Pertsch et al. (2025) or external intervention mod-

ules such as human feedbacks Wang et al. (2024) or multimodal large models for intermediate state evaluation Duan et al. (2024b;a).

To further quantify this robustness, we investigate how the number of external interferences affects task success rates, by extending beyond the single-interference-per-object setting used in our previously reported quantitative results. Specifically, we conduct experiments on all six basic bimanual tasks, focusing on the ID evaluations without loss of generality. We define one interference as a scenario where each object involved in the task is disturbed once. Under this definition, we systematically vary the number of interferences from 0 to 5 and record the average task success rates, which are: 85.0%, 70.0%, 61.7%, 56.7%, 53.3%, and 50.8%, respectively. These results reveal a clear negative correlation between interference frequency and task success rate. However, the rate of decline diminishes as the number of interferences increases, suggesting *a trend of diminishing marginal impact*. One plausible explanation is that as the end-effector gradually approaches the object over time, the spatial freedom available for introducing effective perturbations decreases, thus leading to more stable system performance. Fig. 11 provides illustrative examples of continuous dynamic interference scenarios, and additional results showcasing closed-loop dual-arm control under such conditions can be found in our **supplementary videos**.

Table 6: Quantitative results of VLBiMan's success rates on **four transferred bimanual tasks**.

| Dynamic Interference | Dual-Arm Type | new placements + same objects | | | | | new placements + novel instances | | | | |
|---|---|---|---|---|---|---|---|---|---|---|---|
| | | inserting | unscrew | pouring | reorient | Average Success Rate | inserting | unscrew | pouring | reorient | Average Success Rate |
| No | Contralateral | 18/20 | 16/20 | 17/20 | 15/20 | 82.5% | 17/20 | 14/20 | 14/20 | 14/20 | 73.8% |
| | Humanoid | 19/20 | 17/20 | 16/20 | 15/20 | 83.8% | 18/20 | 16/20 | 13/20 | 14/20 | 76.3% |
| Yes | Contralateral | 13/20 | 15/20 | 15/20 | 12/20 | 68.8% | 11/20 | 12/20 | 11/20 | 10/20 | 55.0% |
| | Humanoid | 14/20 | 15/20 | 14/20 | 13/20 | 70.0% | 12/20 | 13/20 | 12/20 | 10/20 | 58.8% |

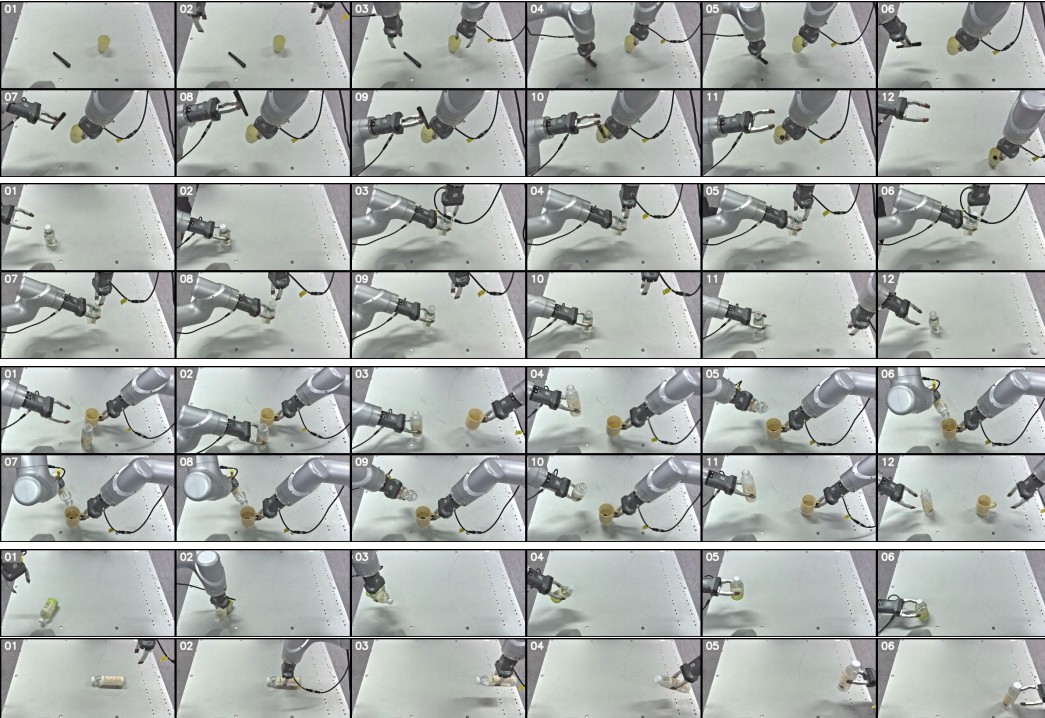

Figure 12: Examples of four transferred bimanual tasks with synchronized dual-arm movement. Segments from top to bottom are tasks `inserting`, `unscrew`, and `pouring`, which have relatively high dual-arm synchronizability. The last row are examples of single-arm task `reorient`, explicitly examining left- or right-handed execution strategies.

## C.4 CROSS-EMBODIMENT TRANSFERABILITY OF VLBIMAN

To further assess the generalization ability of VLBiMan, we investigate its cross-embodiment transferability. Specifically, we evaluate how a one-shot demonstration collected from a human demon-

strator can be transferred to a robotic embodiment with different kinematic and actuation constraints. We report both qualitative visualizations and quantitative results, focusing on four representative bimanual tasks: `inserting`, `unscrew`, `pouring`, and `reorient`, with the corresponding object assets shown on the right side of Fig. 8.

Among them, the `unscrew` and `pouring` tasks preserve the exact same step design and final goals as in the original experiments, thereby serving as direct transfer cases. The `inserting` task, however, introduces embodiment-induced modifications: due to the reduced maximum gripper opening width (from 80 mm to 75 mm), the manipulated cup is no longer placed upside down on the table but instead stands upright. The gripper is required to grasp the cup vertically from the rim and move it to intercept the pen held by the other arm. For the `reorient` task, the manipulated object is replaced with a horizontally placed bottle, and the goal is changed to upright the bottle (with a minimum theoretical rotation of 90 degrees instead of the 180 degrees required when flipping a spoon or spatula). While this can be accomplished by a single arm, we consider a humanoid dual-arm embodiment, as illustrated in the last row of Fig. 6.

As shown in Tab. 6, we further evaluate VLBiMan on the new dual-arm humanoid robot with real-world executions of the four tasks. Following the comparison protocol of Tab. 1, we report success rates on both previously seen objects and novel unseen objects, and additionally record performance under external perturbations. The results demonstrate that, even under a different embodiment, VL-BiMan consistently achieves competitive performance comparable to that on the original dual-arm platform with opposite-side arm installation. This provides convincing evidence of VLBiMan's capability for cross-embodiment transfer and generalization. Fig. 12 presents qualitative visualizations of real-world executions. On this new humanoid platform, we adopt a system configuration where the two arms are maximally synchronized, leading to smoother, more human-like, and more efficient motions. More intuitive dynamic real robot rollouts can be found in our **supplementary videos**.

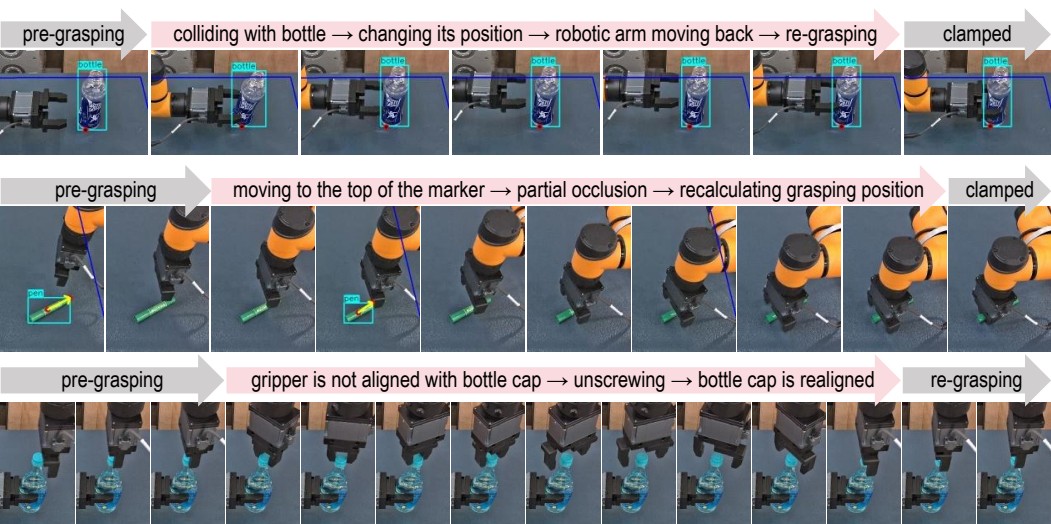

Figure 13: Examples of some interesting findings. *Top row*: this case comes from the pre-grasping phase of `pouring`, where the left arm approaches and grasps the bottle. *Middle row*: this case comes from the pre-grasping phase of `inserting`, where the right arm approaches and grasps the marker from the top direction. *Bottom row*: this case comes from the untwisting bottle cap phase of `unscrew`, where the center of the bottle cap is not aligned with the center point of the end of the gripper. But after a counterclockwise rotation, the upper part of the bottle tilts to the right, and the bottle cap is aligned with the gripper.

## C.5 SOME INTERESTING FINDINGS OF VLBiMan

During our dynamic interference experiments, we observed several interesting and insightful phenomena that further reflect the robustness and adaptability of the proposed VLBiMan framework.

(1) **Dynamic adjustment during initial grasping**: We found that the robot arms often exhibit the ability to dynamically refine their approach trajectories when objects are perturbed just before being

grasped. This can be attributed to the fact that VLBiMan leverages VLMs with strong perception generalization, allowing real-time re-estimation of object poses based on updated visual feedback. Please refer the case in the top row of Fig. 13.

(2) **Continuity despite partial perception failure**: In scenarios where the manipulated object is partially occluded or momentarily not detected (*e.g.*, due to visual obstructions or lighting shifts), the robot can still complete the task. This resilience stems from our modular trajectory composition scheme, which incorporates temporal anchoring of the demonstration-derived trajectory and does not rely on frame-by-frame perfect perception. Please refer the case in the middle row of Fig. 13.

(3) **Tolerance to object displacement during execution**: We also noticed that slight spatial displacements of target objects during intermediate task stages often do not disrupt task execution. This behavior is supported by the task-aware decomposition and image-moment-based orientation extraction modules, which are both designed to operate on robust and low-frequency visual features (*e.g.*, binary masks), making the entire system less sensitive to minor deviations and noise. These findings collectively highlight how VLBiMan benefits from the synergy between robust perception modules and structured motion control, leading to more fault-tolerant and adaptable bimanual manipulation. Please refer the case in the bottom row of Fig. 13.

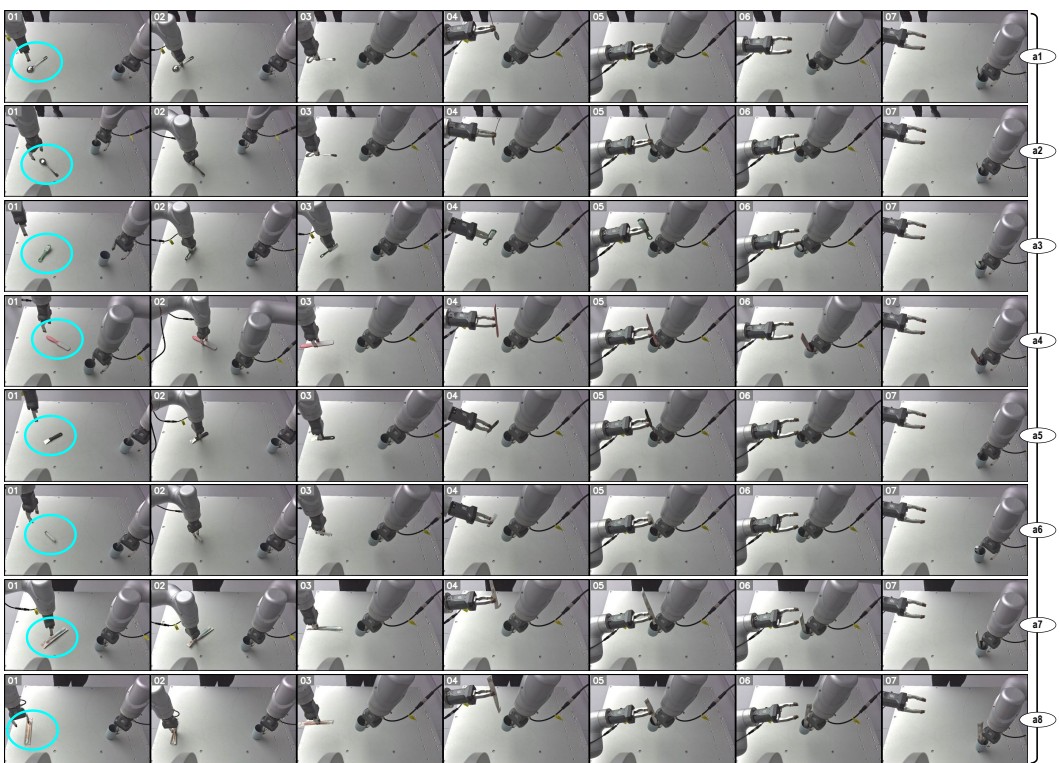

Figure 14: Taking the `inserting` task as an example, we replaced the marker pen held by the left arm with other rectangular objects that were completely different (including *spoon* in cases **a1/a2**, *brush* in cases **a3/a4**, *spatula* in case **a5**, *syringe* in case **a6**, and toothbrush in cases **a7/a8**). These newly added objects are circled in cyan color. We found that VLBiMan could still successfully locate the objects based on our designed method of using the centroid of the object's 2D mask as a representative point. Furthermore, it accurately estimated the object's pose using the orientation estimation method in Algorithm 1, thereby helping to stably grasp these objects and ultimately achieve the inserting task objective.

## C.6  DISCUSSION OF HUMAN-IN-THE-LOOP REFINEMENT

While VLBiMan is designed as a training-free and highly automated pipeline, the initial **task-aware spatio-temporal decomposition** may occasionally require minor human refinement during its *first-time execution on a new task*. These refinements primarily concern safety and robustness adjustments that are difficult to infer from a single demonstration alone. Typical examples include veri-

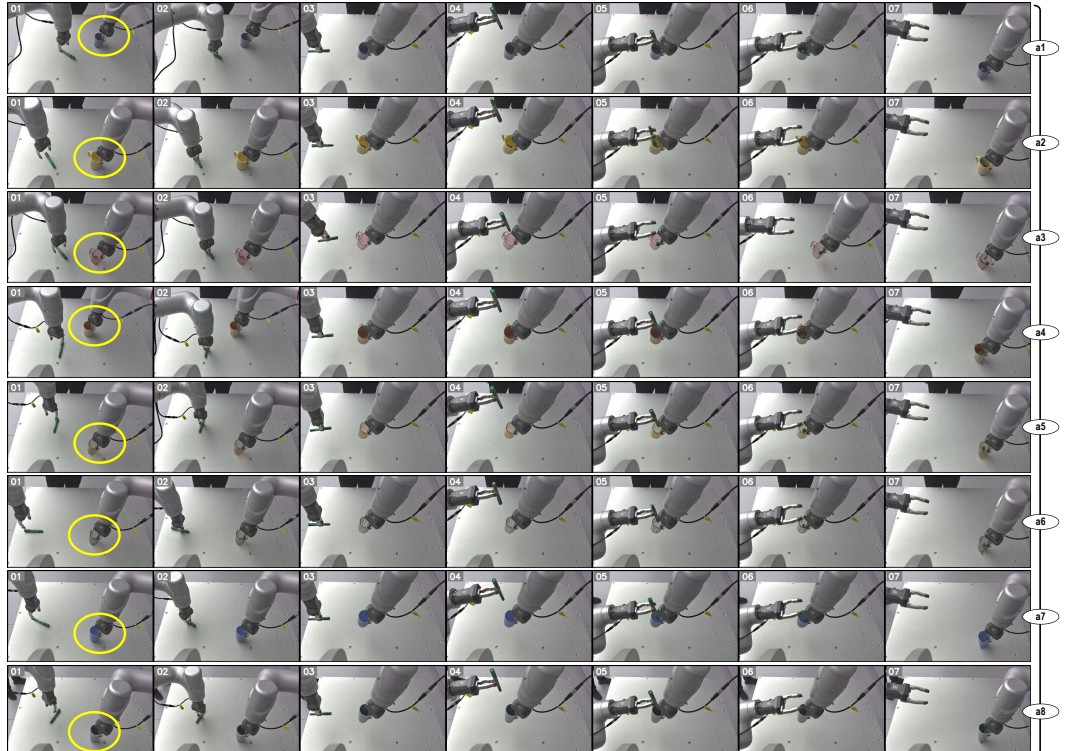

Figure 15: Taking the `inserting` task as an example again, we changed the cup being grasped by the right arm to cups of completely different shapes (including *mugs with handles* in examples **a1/a2/a3/a4**, and *ordinary cups without handles* in examples **a5/a6/a7/a8**). These newly added objects are circled in yellow color. We found that VLBiMan can still successfully locate the objects using our designed method of using the foremost point of contact between the object and the table as a representative point (note that at this time, it is not necessary to use the orientation estimation method to estimate the object's pose again), thus helping to stably grasp these cups and ultimately complete the inserting task objective.

fying the tilt angle when grasping a mug's handle or ensuring that the downward orientation of the right arm during unscrewing avoids exerting lateral pressure on deformable bottles.

Importantly, such refinements occur **only once per task**, at decomposition time, and do not reappear during any subsequent executions. Once the primitive boundaries and key waypoints are validated, VLBiMan entirely relies on (1) VLM-based spatial adaptation and (2) trajectory composition to handle object pose variation, shape diversity, and long-horizon skill chaining. In practice, we find that only a small subset of tasks require any refinement at all, and the operator does not need to possess expert-level manipulation knowledge.

We acknowledge that fully automatic and reliable segmentation remains an open challenge in few-shot imitation learning. Hardware-assisted demonstration capture (e.g., instrumented gloves or hand-held trackers) could offer increased precision, though such solutions incur embodiment mismatch, reduced dexterity, and alignment overhead. Exploring more principled automatic segmentation approaches while preserving usability remains an important future direction.

## C.7 ROBUSTNESS OF OBJECT REPRESENTING POINTS

A core design choice in VLBiMan is the use of simple yet highly generalizable object representing points, which serve as anchors for both task-aware decomposition and cross-object adaptation. In practice, we adopt either the center of the object's 2D mask or the foremost contact point between the object and the supporting surface. Despite their simplicity, these representations prove surprisingly robust across a wide variety of object geometries. Because these points are derived directly from VLM-assisted object segmentation, they require no object-specific modeling like the widely-used 6D object pose estimation Lin et al. (2024); Wen et al. (2024a), and naturally extend to unseen

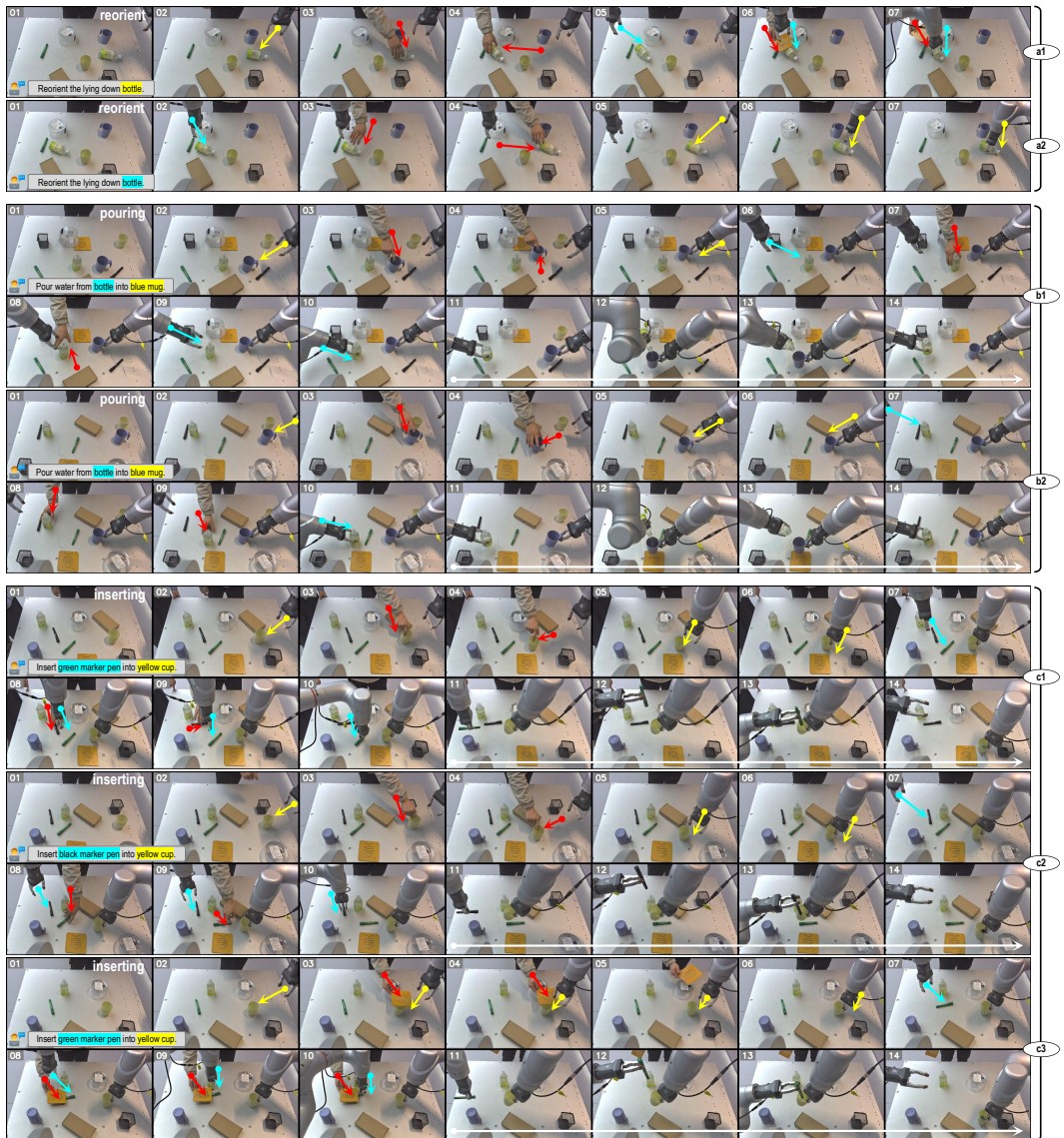

Figure 16: Visualization of test results for VLBiMan's robustness performance in cluttered scenarios. We selected three tasks, `reorient` (corresponding to examples **a1/a2**), `pouring` (corresponding to examples **b1/b2**), and `inserting` (corresponding to examples **c1/c2/c3**), for extensive evaluation. In these examples, there are not only various irrelevant objects that can easily lead to **semantic ambiguity** and **execution obstacles**, but we will also **unexpectedly rearrange** the target object during the pre-grasping stage. This requires the VLBiMan to be able to quickly and nimbly find the target object again from the cluttered scene based on the task's linguistic instructions. This process faces many significant non-trivial challenges. The red arrow indicates the direction of the manually moved or deliberately obscured object (interfering). The cyan arrow and yellow arrow indicate the movement direction of the left and right robotic arms (chasing) respectively.

objects with distinct shapes, sizes, and surface profiles. This choice also provides a robust abstraction that generalizes across intra-class variations, imperfect geometry, and partial occlusions. And the system adapts by reattaching the invariant primitive to newly inferred representing points, maintaining functional consistency even in scenarios where popular 6D pose methods Lin et al. (2024); Wen et al. (2024a) tend to fail due to symmetry or texture sparsity.

To further validate this robustness, we conducted additional experiments in the `inserting` task, where geometric variations and small pose offsets are particularly challenging. And the used hardware and platform are the dual-arm humanoid robot (refer Fig. 8). As shown in Fig. 14 and Fig. 15,

the results consistently show that these lightweight representations support reliable cross-instance transfer without re-tuning, enabling accurate alignment even when objects differ significantly from those used in the original demonstration (e.g., varying mug/cup shapes or cuboid object's sizes). These findings highlight that VLBiMan's adaptation does not depend on high-fidelity 3D reconstruction or complex shape descriptors. Instead, object-centric points extracted from 2D perception are sufficient to drive effective and scalable bimanual manipulation. For all examples in Fig. 14 and Fig. 15, we have provided corresponding real-robot rollout videos in the **supplementary materials**, and continue to support the application of perturbation to these entirely new categories of objects during the initial grasping phase, further demonstrating the strong generalization and wide versatility of VLBiMan.

## C.8 Testing VLBiMan under Cluttered Scenarios

To further examine VLBiMan's robustness to complex perceptual conditions and more abstract natural language descriptions, we conduct additional experiments in cluttered tabletop environments. These scenes contain at least five distractor objects whose categories, shapes, or colors resemble the target object, increasing semantic ambiguity and spatial interference. Using the same pipeline as in the main paper (without modifying any module), we evaluate `reorient`, `pouring`, and `inserting` tasks under distractors, dynamic object relocation, and partial occlusion. We still utilized the dual-arm humanoid robot (refer Fig. 8) as the hardware and platform. In these clutter tests, the VLM module must rely solely on the language instruction to identify the correct target and provide a stable grounding for subsequent geometric adaptation.

As shown in Fig. 16, across all cluttered configurations, VLBiMan consistently identifies the appropriate object and completes the tasks with high reliability. Even when the object is **perturbed mid-execution** or intentionally **partially obscured**, the system re-aligns the representing points and resumes the correct trajectory within a single perception–planning cycle ($\sim$1 second). Corresponding visual results along with various indicating arrows are provided in Fig. 16. To our knowledge, many of the challenges in these examples lack systematic exploration in the current field of robotic manipulation. For instance, even when the target object is partially obscured during manipulation, VLBiMan can still locate the target and execute the grasping action accurately (see examples **a1/c3**). When there are multiple selectable target objects in the scene, VLBiMan can consistently eliminate ambiguous interference from very similar objects (in examples **b1** and **b2**, where both require grasping the blue mug, the system will not grasp the handleless yellow cup. And in examples **c1** and **c2**, where the system needs to grasp the green and black marker pens respectively, it will not mistakenly grasp the other unwanted marker pen).

To sum up, these new experiments further validate that VLBiMan extends beyond template verb-conditioned tasks and remains robust under linguistic variation, distractor-rich scenes, and environmental disturbances. We highly recommend watching our recorded rollout videos provided in the **supplementary materials** to get a more intuitive feel for VLBiMan's stunning performance.

## C.9 Ablation Studies of the Interpolation Density

During the pre-grasp phase in each task, VLBiMan introduces a set of interpolated waypoints parameterized by an interpolation density $n$. This design serves two purposes: (1) ensuring a *smooth and safe approach trajectory* that reduces the risk of prematurely colliding with the object, and (2) helping to *improve robustness against external disturbances*. Without interpolation, the end-effector may directly execute a long straight-line motion from its initial configuration toward the demonstration-aligned grasp pose, which increases the chance of accidental contact, especially when the object has been shifted or rotated. Here we discuss how to find the optimal value of $n$.

As shown in Tab. 7, our ablation on the choice of $n$ reveals clear benefits: *higher interpolation density leads to improved stability under perturbations*, including cases where the object is intentionally repositioned by a human or slightly displaced by the robot's own motion during execution. The gradual, multi-step approach allows the controller to continually re-evaluate object-relative anchors and correct small deviations on the fly. Notably, we find diminishing returns beyond a moderate range of $n$ (e.g., relatively small values), indicating that the pre-grasp interpolation strategy does not rely on excessive tuning. Overall, these studies demonstrate that a carefully selected number of interpolated points contributes to both safety and disturbance resilience, enhancing VLBiMan's reliability

Table 7: Ablation experiments of the interpolation density $n$. We utilized the dual-arm humanoid robot platform to conduct four bimanual manipulation tasks. Similar to Tab. 6, we still divided them into objects that appeared in the single demonstration and new objects that did not appear in the demonstration. Each task under each setting was executed with 20 trails, and the average success rate was calculated. To ensure reliable searching of the optimal $n$, we did not add any additional dynamic interference in each trail, and stopped the task immediately after the initial grasping stage finished or failed of the test (indicating the **pre-grasping** only performance).

| Interpolation Density $n$ | new placements + same objects | | | | | new placements + novel instances | | | | |
|---|---|---|---|---|---|---|---|---|---|---|
| | inserting | unscrew | pouring | reorient | Average Success Rate | inserting | unscrew | pouring | reorient | Average Success Rate |
| $n = 3$ | 15/20 | 14/20 | 12/20 | 13/20 | 67.5% | 13/20 | 12/20 | 11/20 | 11/20 | 58.8% |
| $n = 4$ | 18/20 | 15/20 | 15/20 | 16/20 | 80.0% | 17/20 | 14/20 | 14/20 | 14/20 | 73.8% |
| $n = 5$ | 19/20 | 17/20 | 18/20 | 16/20 | 87.5% | 18/20 | 16/20 | 17/20 | 15/20 | 82.5% |
| $n = 6$ | 19/20 | 19/20 | 18/20 | 17/20 | 91.3% | 19/20 | 18/20 | 17/20 | 16/20 | **87.5%** |
| $n = 7$ | 19/20 | 18/20 | 18/20 | 18/20 | **92.5%** | 18/20 | 19/20 | 17/20 | 16/20 | **87.5%** |
| $n = 8$ | 19/20 | 19/20 | 17/20 | 18/20 | **92.5%** | 19/20 | 18/20 | 16/20 | 17/20 | **87.5%** |

in real-world deployments. Practical deployments can adopt a medium density ($n$=6) that *balances pre-grasping accuracy and computational efficiency*, as used in our main experiments.

## D    STATEMENT ON THE USE OF LARGE LANGUAGE MODELS

During the preparation of this manuscript, we used the ChatGPT language model **exclusively for linguistic refinement**, including grammar correction and stylistic improvement. The model did not contribute to research design, methodology, experiments, or analysis. All scientific content and intellectual contributions are solely the work of the authors.

