# OpenReview forum: "VLBiMan: Vision-Language Anchored One-Shot Demonstration Enables Generalizable Bimanual Robotic Manipulation"
_ICLR.cc/2026/Conference — ICLR 2026 Poster_

### Official Review · Reviewer_DSMU · 2025-10-27

**Soundness:** 3
**Presentation:** 3
**Contribution:** 3
**Rating:** 8
**Confidence:** 4

**Summary:**

This paper introduces VLBiMan, a framework for generalizable bimanual robotic manipulation using only a single human demonstration. The method decomposes the demonstration into reusable motion primitives and adapts them to novel objects or environments through vision-language models (VLMs) and geometric reasoning. Unlike imitation learning approaches that rely on large datasets, VLBiMan leverages semantic grounding and geometric anchors to perform new tasks without retraining. The authors demonstrate strong results on 10 real-world bimanual manipulation tasks including tool use and long-horizon skill compositions, achieving significantly higher success rates than baselines.

**Strengths:**

- Modular design: The decomposition–adaptation–composition pipeline is intuitive and well-grounded, allowing for task-aware reuse of motion primitives.
- One-shot generalization: Demonstrating adaptation to new objects, poses, and even robotic embodiments is impressive.
- Comprehensive empirical evaluation: Ten tasks, multiple baselines, both static and dynamic settings, and thorough ablations (Table 3) provide convincing evidence.
- Clarity and presentation: Figures clearly illustrate both conceptual flow and empirical outcomes. The writing is clear and professional.

**Weaknesses:**

- Limited novelty in underlying modules: While the integration is well-engineered, the components are largely standard.
- Dependence on heuristics and manual refinement: The segmentation step requires human-in-the-loop refinement, which undermines full automation and scalability.
- Lack of comparison with recent closed-loop VLA policies: The baselines focus on modular imitation systems (ReKep, Robot-ABC), but not large VLA models.

**Questions:**

- How sensitive is VLBiMan to segmentation quality in the one-shot demonstration?
- Can the approach scale to more abstract task descriptions beyond manipulation verbs?
- Could the authors provide computational efficiency details?

---

> ### Author Response · Authors · 2025-11-26
> **Response by Authors to Reviewer DSMU [Part 1]**
>
> ## General Response
>
> We sincerely thank the reviewer for the careful reading, constructive feedback, and very positive overall assessment of our work. We truly appreciate your high recognition of VLBiMan's modular design, one-shot generalization capability, strong empirical results, and clear presentation. Your encouraging evaluation and "accept" recommendation mean a great deal to us. We address each of your concerns in detail below and hope our clarifications will further strengthen your confidence in the contribution.
>
> Here, we briefly reiterate the main advantages of VLBiMan that align with your observations: **(1)** Modular, task-aware pipeline enabling interpretable decomposition–adaptation–composition. **(2)** Strong one-shot generalization to new objects, poses, and embodiments through vision-language anchoring. **(3)** Comprehensive real-world validation across 10 challenging bimanual tasks with robust performance. **(4)** Practical, training-free deployment without collecting large datasets or retraining policies.
>
> We thank the reviewer again for the insightful comments and the enthusiasm shown toward our work. And we hope the detailed responses in the following sections will further clarify and reinforce the strengths and novelty of our work.
>
> ***
>
> ## W1: On the concern regarding limited novelty of individual modules
>
> We appreciate the reviewer's perspective and would like to clarify that, although the three modules in VLBiMan are intentionally lightweight, the **overall problem-solving paradigm** is indeed novel and impactful for scalable bimanual manipulation. Specifically:
>
> > (1) **Task-aware decomposition using VLMs from a single bimanual demonstration** is fundamentally different from prevailing end-to-end VLA pipelines built on large teleoperation datasets. We intentionally demonstrate that imitation learning need not rely solely on large VLA policies—*leveraging human priors through semantic and geometric abstraction is itself an effective and under-explored paradigm*.
>
> > (2) **Selective and semantics-guided perception**, instead of universal 6D pose estimation or grasp-pose prediction, allows VLBiMan to rely on highly robust VLM/SAM2-based grounding and to adaptively decide whether orientation estimation is needed. This design yields far more reliable generalization across heterogeneous household objects.
>
> > (3) **Reusing decomposed invariant motion primitives as reusable atomic skills** maximizes data efficiency and enables long-horizon compositions. We demonstrate only 10 tasks, but VLBiMan can in principle scale to many more without collecting new demonstrations.
>
> Finally, we note that a recent **Science Robotics 2025** article (titled *"Learning a Thousand Tasks in a Day"* with the link https://www.science.org/doi/10.1126/scirobotics.adv7594) also proposes decomposing demonstrations into alignment/interaction phases to improve generalization, which shares the similar motivations with us and provides further independent evidence for the conceptual validity and timeliness of VLBiMan's single-shot decomposition paradigm.
>
> ## W2: On the dependence on human-in-the-loop refinement
>
> Thank you for pointing this out. We acknowledge that the first-stage task-aware decomposition occasionally requires minimal human refinement (e.g., adjusting grasp tilt for mugs, ensuring vertical alignment during unscrewing). These refinements are **task-level safety/robustness adjustments**. They appear only during the first-time decomposition of each task, and do not recur during all subsequent executions, unless new long-horizon skills are being composed.
>
> In practice, only a few tasks require such refinements, and the operator need not be an expert. After decomposition is finalized once, VLBiMan relies solely on *VLM-based adaptation and trajectory composition*, which eliminates any human dependency during deployment.
>
> We agree that fully automated spatio-temporal segmentation is an important future direction. Hardware-assisted demonstrations (e.g., instrumented gloves as in DexCap or hand-held end-effectors as in UMI/DexUMI) offer potential solutions but introduce their own mismatch and embodiment challenges. A detailed discussion has been added to `Appendix C.6`.

---

> ### Author Response · Authors · 2025-11-26
> **Response by Authors to Reviewer DSMU [Part 2]**
>
> ## W3: On the lack of comparison with closed-loop VLA policies
>
> We appreciate this question and fully agree it is important to contextualize VLBiMan relative to recent VLA policies. Our choice of baselines follows two principles:
>
> > (1) **Fairness**: closed-loop VLA policies such as ACT or Diffusion-Policy require task-specific datasets (often tens to hundreds of teleoperation trajectories). Their success rate scales directly with data quantity and model size, making them incomparable to our **training-free paradigm**. For fairness, we therefore compare to training-free one-shot or modular methods (Mechanisms, MAGIC, ReKep, Robot-ABC), which share at least one conceptual axis with VLBiMan.
>
> > (2) **Empirical attempts**: we attempted to deploy large pretrained VLA models such as **$\pi_0$** directly on our hardware setup and tasks, but they consistently failed **under embodiment mismatch**. Finetuning them would again require collecting significant task-specific data, defeating the purpose of our paradigm and making a fair comparison impossible.
>
> We emphasize that our evaluation goal is not to outperform powerful data-driven policies, but to demonstrate what **training-free, single-demonstration** generalization can achieve in complex bimanual manipulation.
>
> ## Q1: Sensitivity to segmentation quality
>
> As noted in `W2`, segmentation is performed **only once per task**, and once validated, decomposition is fixed thereafter. We currently lack a principled metric for evaluating segmentation granularity, and exhaustively exploring all possible decompositions is computationally impractical. Empirically, however, we observe:
>
> > (1) VLBiMan's downstream robustness is **not strongly sensitive** to the specific segmentation boundaries as long as the resulting execution is valid.
>
> > (2) Decomposition quality is **not a dominant factor** affecting the final task's success rate.
>
> Thus, while improving automated segmentation is valuable future work, it is not a key bottleneck in VLBiMan's practical performance.
>
> ## Q2: Scaling to more abstract language descriptions
>
> This is indeed an insightful direction. To explore it, we conducted **new cluttered-scene experiments** (`Appendix C.8`) where VLBiMan must interpret diverse language descriptions, disambiguate among multiple distractor objects, and maintain correct grounding under motion or partial occlusion.
>
> We evaluated *pouring/inserting/reorient* tasks with ≥5 distractor objects whose shapes or semantics closely resemble the target. Using only language instructions and VLM-based grounding, VLBiMan consistently identified the correct target and completed the tasks, even with external perturbations. Videos are provided in `new_exps_clutter_test.mp4`.
>
> These results suggest that VLBiMan can indeed handle richer, non-template language instructions beyond simple manipulation verbs.
>
> ## Q3: Computational efficiency
>
> VLBiMan is highly efficient and lightweight. Most modules run entirely on CPU. Only Florence-2 and SAM2 require a consumer-grade GPU. By resizing inputs to 720-px on the long side, the GPU memory usage remains around **8 GB**. The VLM/VFM inference takes about **0.6 seconds**, and the full perceive–plan–execute cycle completes within **~1 second**. This enables responsive real-time execution, as reflected in our provided rollout videos. We will release code after de-anonymization to further support reproducibility.
>
> ***
>
> ## Closing Remark
>
> We once again thank the reviewer for the thoughtful evaluation and encouraging assessment. We hope that the detailed clarifications provided above address all concerns and further highlight the conceptual novelty, practical utility, and empirical strength of VLBiMan. We greatly appreciate the reviewer's interest in our work and will incorporate these suggestions to further improve the final version.

---

### Official Review · Reviewer_ueUa · 2025-10-28

**Soundness:** 4
**Presentation:** 3
**Contribution:** 3
**Rating:** 4
**Confidence:** 4

**Summary:**

VLBiMan can be directly deployed in a new environment without the need for further training, simply by using RGB images from a single human dual-arm demonstration, 6Dof poses of the robotic arm, gripper switch states, and natural language task descriptions. The method performed approximately 4,200 real robot rollouts on 10 self-built dual-arm tasks, with an average success rate of 78% (ID) / 59% (OOD), significantly outperforming 5 strong baselines, and demonstrated robustness across embodiment migration, lighting, and continuous disturbances. However, the paper has some missing details in the implementation of the method. At the same time, it does not conduct further discussion and analysis on failed cases, nor does it carry out statistical analysis such as confidence level on the main experimental results. There is room for further improvement.

**Strengths:**

1. There is no need to re-collect data or fine-tune the model for new objects or new robots, and sufficient experimental verification has been conducted at the same time.
2. Through the modular design concept, the "unchanging skills" and the "variables to be adapted" are explicitly separated. Combined with VLM for semantic segmentation and geometric offset calculation, a lightweight and interpretable process has been achieved.
3. A progressive inverse kinematics solution and proximal - vertical collision compensation method were proposed, effectively alleviating conflicts in time and space, supporting synchronous/asynchronous hybrid execution, and achieving coordination of dual-arm operations.
4. The same demonstration can be directly deployed to the opposed dual-arm platform and the humanoid dual-arm platform. The success rate of the experimental test still remains at around 70%, proving the universality of the method on different robot platforms.
5. Under conditions such as illumination changes and continuous external dynamic disturbances, it still maintained a high success rate. The experimental evaluation was comprehensive, demonstrating the robustness of VLBiMan.

**Weaknesses:**

1. The experimental analysis lacks a thorough examination of key hyperparameters, including the geometric equivalence threshold ε_g, the stability window ε, and the interpolation density n=6. These parameters directly influence the quality of decomposition and trajectory generation; however, the manuscript does not provide justification for their selected values or the rationale behind the search ranges explored.
2. The criteria for selecting representative points remain ambiguous. Specifically, the paper fails to clarify under what conditions the mask centroid is chosen versus planar contact points, and it does not detail the procedure for identifying planar contact points. This lack of transparency may hinder the method’s reproducibility and limit its practical utility in engineering applications.
3. The proposed method may face challenges in accurately estimating the orientation of symmetric objects. For example, Algorithm 1 relies primarily on the principal axis derived from the second moment, which introduces a 0°–180° rotational ambiguity for rotationally symmetric or near-circular objects. The paper does not address whether this ambiguity could lead to functional failure, nor does it incorporate strategies—such as leveraging vision-language model (VLM) semantics—to resolve such ambiguities.
4.Language is used only to generate object prompts, without incorporating a language-action alignment loss or conducting ablation studies on the influence of different linguistic formulations on segmentation performance. As a result, the integration of the language modality appears superficial, and the potential benefits of multimodal reasoning within the framework are not fully realized or demonstrated.
5. All reported results are presented solely as average success rates, without accompanying measures of statistical variability such as standard deviations, confidence intervals, or significance testing. Given the limited number of trials (e.g., 20), differences such as a 78% versus 70% success rate may not be statistically significant. Furthermore, the absence of detailed failure case analyses prevents a comprehensive understanding of critical issues such as spatio-temporal conflicts.

**Questions:**

1. With regard to the sensitivity of key hyperparameters, could you please provide sensitivity analysis curves or ablation study tables for critical parameters such as ε_g, ε, and n? Additionally, could you clarify whether these hyperparameters are consistently set across different tasks?
2. Concerning the potential limitations in estimating orientations for symmetric objects, particularly rotationally symmetric ones such as bottle caps and cup rims, could orientation ambiguity lead to task failure? If so, what proportion of failures can be attributed to this issue, and are there any proposed solutions to mitigate such failures?
3. Regarding the lack of clarity in explaining the strategy for selecting representative points, could you offer further elaboration on the criteria used to determine when to employ the center of gravity versus planar contact points?
4. With respect to the insufficient analysis of failure cases, in scenarios involving spatio-temporal conflicts during bimanual operations, how is collision detected in real time? Were any statistical analyses performed on the observed failure cases? If available, could you provide quantitative data on the frequency and types of failures encountered in the experiments?
5. Regarding the design of ablation studies, is there a plan to conduct ablation experiments concerning language inputs in future work? Specifically, if attribute descriptors (e.g., color, shape) in the task instructions are removed or altered, how would this affect the VLM segmentation recall rate and the overall task success rate?
6. Concerning the absence of statistical significance testing, could you please include 95% confidence intervals or bootstrap confidence intervals in the main results table? Furthermore, could a paired two-sample statistical test be applied to assess whether the performance differences between VLBiMan and other baseline methods are statistically significant (p < 0.05)?

---

> ### Author Response · Authors · 2025-11-26
> **Response by Authors to Reviewer ueUa [Part 1]**
>
> ## General response and appreciation
>
> We sincerely thank the reviewer for the careful reading and the constructive feedback. We are grateful for the reviewer's recognition of VLBiMan's practical deployability without retraining, the modular decomposition that separates invariant primitives and variable components, the proposed kinematic / compensation solutions for coordination, and the empirical evidence of cross-embodiment transfer and robustness.
>
> Your detailed points about hyperparameter sensitivity, representative-point selection, symmetry ambiguity, deeper multimodal use of language, and statistical rigor are all well taken. We fully agree these are important for clarifying the method and improving reproducibility. Below we firstly give a short reiteration of the proposed VLBiMan, then we respond to each weakness and question in turn and summarize concrete additions we will make in the revised manuscript and supplementary material.
>
> ## Brief reiteration of VLBiMan's core strengths
>
> > (1) **One-shot, training-free generalization**: decompose a single demonstration into reusable atomic skills and adapt only the variable components via VLM-anchoring.
>
> > (2) **Practical modularity**: task-aware decomposition + VLM-based perception + trajectory composition yields an interpretable, deployable pipeline.
>
> > (3) **Empirical robustness and transfer**: extensive real-robot tests (10 tasks, many rollouts) and cross-embodiment experiments illustrate practical applicability.
>
> ***
>
> ## Q1 & W1: Sensitivity of hyperparameters ($\epsilon_g$, $\epsilon$, $n$)
>
> Thank you for pointing out the need for clearer explanations and supporting experiments regarding hyperparameter sensitivity. Below we clarify each parameter's role:
>
> > ### (1) Geometric equivalence threshold $\epsilon_g$
> This threshold (used by `Eq. (2)`) is **not required for all tasks**, and its value depends on the specific objects involved in the task's decomposed segments. It is an **experience-based constant**, applied only during task decomposition once, and reused across all scenes of the same task. Its purpose is mainly **expository**: to express the definition of Atomic Skills using rigorous mathematical relations that distinguish contact vs. non-contact segments and further divide adaptable vs. invariant segments.
>
> > ### (2) Stability window $\epsilon$,
> Defined in `Appendix B.2`, this threshold detects whether an object has moved unexpectedly (e.g., human disturbance or slight premature contact). We set $\epsilon$ as 10 mm for all experiments. Values such as 5 mm or 15 mm also work. Changing $\epsilon$ only affects **when replanning is triggered**, and does not influence algorithms downstream. Therefore, a dedicated ablation is unnecessary.
>
> > ### (3) Interpolation density $n$
> Used in `Eq. (4)` to discretize pre-grasp trajectories, enabling reactive collision avoidance and disturbance handling. A single value **$n$ = 6 works across all tasks and all objects**. We agree with the reviewer that experimental justification is needed. Therefore, we added new sensitivity experiments on four tasks (*inserting / unscrew / pouring / reorient*) by varying $n$ and measuring the average pre-grasp success rate (20 trials per setting). The results are summarized in `Appendix C.9` and `Table 7`, confirming that $n$ = 6 lies near the empirical optimum.
>
> We hope these clarifications and new experiments improve transparency and reproducibility.
>
> ## Q2 & W3: Orientation ambiguity for symmetric objects
>
> We fully agree that rotational symmetry introduces 0°–180° ambiguity when using principal-axis estimation from 2D masks. Our treatment is as follows:
>
> > ### (1) Which objects require orientation estimation?
> Only objects whose **functional grasp direction depends on orientation** use `Algorithm 1`, such as marker pen, spoon, spatula, and bottles lying horizontally.
>
> > ### (2) Which objects do not require orientation estimation?
> Highly symmetric or circular objects (e.g., standing bottles, upright cups, bowls) do not need orientation estimation. Their functional grasps are **implicitly embedded** in the one-shot demonstration and remain consistent during deployment.
>
> For examples, in the *pouring* task, the bottle is always upright. The gripper approaches orthogonally to its main axis. In the *tool-use scoop* task, the bowl is always upright. Only its top-center point needs to be localized. Thus, VLBiMan is **not affected** by orientation ambiguity for these objects. To make this explicit, we added an updated `Appendix Table 4` listing, for each task, which objects require orientation estimation and which do not.

---

> ### Author Response · Authors · 2025-11-26
> **Response by Authors to Reviewer ueUa [Part 2]**
>
> ## Q3 & W2: Criteria for selecting representative points
>
> Thank you for appreciating the core idea behind our abstraction of representative points. We agree that this deserves deeper analysis. We have proposed two types of representative points. As shown in `Fig. 3`, we use the mask centroid or front-most planar contact point with the table to represent an object. This simplification works consistently across all experiments.
>
> To further validate the generality of this representation, we conducted **new robustness experiments** in the *inserting* task involving: **(1)** different long/rod-like objects (spoon, brush, spatula, eraser, syringe, etc.) represented by mask centroids; **(2)** diverse mugs and containers (varied geometry and partial occlusion) represented by planar contact points.
>
> Despite occasional errors, VLBiMan performed robustly across these diverse shapes and occlusion conditions. All results are reported in `Appendix C.7` and included in the supplementary video (`new_exps_rep_pts_test.mp4`). We also summarized all representative-point choices per object across all tasks in the updated `Appendix Table 4`.
>
> ## Q4 & W5: Failure case analysis and spatio-temporal conflicts
>
> We appreciate your insightful question. In fact, a detailed failure breakdown was already provided in `Sec. 4.3` (`Table 3`; `Fig. 7`). Below is a recap and further clarification.
>
> For the **failure types distribution** (from `Fig. 7`), they ordered by frequency include Initial Grasp Executing (45%), Dual-Arm Coordination (21%), Initial Grasp Computing (12%), VLM Perception (6%), and Trajectory Optimization (6%). More discussion of these failure cases appears in `Sec. 4.3`, second paragraph.
>
> As for the **spatio-temporal conflicts**, the reviewer's concern corresponds to Dual-Arm Coordination, which accounts for ~21% of failures and is not the majority. Actually, conflicts are largely prevented **before execution** by: **(1)** detection of excessively close left/right arm waypoints during one-shot decomposition; **(2)** human-in-the-loop refinement to remove risky motions; **(3)** progressive IK and vertical compensation during execution. This design is intentionally lightweight yet highly effective.
>
> ## Q5 & W4: Influence of varied language inputs
>
> We appreciate your interest in this direction (and other two reviewers `nQo1` and `uWbN` also raised similar questions). To address it, we conducted **new robustness experiments on cluttered scenes** requiring strong language-based disambiguation.
>
> We choose three tasks including *pouring*, *inserting*, and *reorient* for further testing. The platform is the humanoid dual-arm robot (refer `Fig. 8`). We added ≥5 distractor objects per scene (similar categories, shapes, and sizes). These setups require the VLM to interpret language input and precisely localize the correct target objects. Besides, we still included mid-execution disturbances such as object relocation and partial occlusion.
>
> Finally, results showed that VLBiMan consistently succeeded in attending to the correct object and completing the tasks. Full quantitative results are reported in `Appendix C.8`, with videos included (`new_exps_clutter_test.mp4`). We hope this alleviates the reviewer's concern about VLBiMan's robustness under varied linguistic formulations.
>
> ## Q6 & W5: Absence of statistical significance testing
>
> Thank you for raising this important point. We follow the common embodied-robotics evaluation paradigm: ≥20 trials per task, using success rate as the primary metric. This is consistent with leading works whose trial counts usually fall between 10–20.
>
> Although computing variance or confidence interval for real-world experiments is nonstandard in the robot manipulation field, we fully understand the reviewer's concern. Therefore, we increased trials **from 20 to 25** for the main experiments during rebuttal. Updated `Tables 1/2` reflect these new statistics. And the success rates remain stable and VLBiMan maintains its strong advantage. Due to time constraints, we could not repeat all baselines across all tasks with more trials, but the increased sample size improves the reliability of reported numbers.
>
> ***
>
> ## Final Remarks
>
> We truly appreciate your constructive comments, which helped us significantly refine the clarity, rigor, and completeness of the manuscript. We have added new analyses, expanded experiments, clearer explanations, and updated tables as suggested. We hope our responses address your concerns and further demonstrate the robustness, transparency, and technical soundness of VLBiMan. Thank you again for your valuable feedback.

---

### Official Review · Reviewer_uWbN · 2025-10-30

**Soundness:** 3
**Presentation:** 3
**Contribution:** 3
**Rating:** 6
**Confidence:** 3

**Summary:**

This paper introduces VLBiMan, a framework for generalizable bimanual manipulation that learns reusable skills from a single human demonstration by decomposing tasks into invariant primitives and dynamically adapting variable components via vision-language grounding. This approach eliminates the need for extensive demonstration datasets and policy retraining, enabling robust adaptation to scene ambiguities, novel objects, and disturbances. The system achieves compositional generalization by splicing atomic skills for long-horizon tasks and demonstrates strong cross-embodiment transfer, allowing skills learned from human demo to be directly deployed on different robotic platforms, thereby advancing versatile dual-arm manipulation in unstructured environments.

**Strengths:**

1. This paper achieves remarkable data efficiency and generalization by learning reusable skills from a single demonstration through task-aware decomposition and vision-language adaptation.
2. The experiments conduct detailed analysis on the modular system design and system robustness, including ablations on different modules and breakdowns of failure cases.
3. The paper is well-written, presenting a complex technical system with conceptual clarity and a logical narrative that is easy to follow.

**Weaknesses:**

1. The scenes involved in the paper are clean. The paper does not show the cases where some distracted objects are placed in cluttered scenarios, which may pose some challenges to the capabilities of the visual-language anchored adaptation modules.

**Questions:**

1. How much human efforts are required in the "human-in-the-loop refinement" in the Spatiotemporal Segmentation?
2. I'm wondering how your pose estimation method (proposed in 3.2) compares to off-the-shelf 6D pose estimation methods, especially for some symmetrical objects.
3. How do you deal with potential collision during each motion besides grasp approach?

---

> ### Author Response · Authors · 2025-11-26
> **Response by Authors to Reviewer uWbN [Part 1]**
>
> ## General Response.
> We sincerely thank the reviewer for the careful reading of our paper and for the encouraging assessment of VLBiMan. We greatly appreciate your positive remarks on the data efficiency of our one-shot skill extraction, the clarity of our modular system design, and the thoroughness of our experimental analyses. Your acknowledgement of both the conceptual narrative and the practical advantages means a great deal to us, especially in terms of robustness, skill reuse, and cross-embodiment transfer.
>
> We fully agree that understanding the limits of vision-language–anchored adaptation and evaluating the system under more challenging real-world conditions are essential. We therefore address your concerns and questions in detail below, and we hope our clarifications and additional analyses further illustrate the intended scope and contributions of VLBiMan.
>
> ***
>
> ## W1: Concern about cluttered scenarios
>
> Thank you for pointing out this important limitation. We agree that the main paper primarily presents experiments in relatively clean scenes, which could obscure the true robustness of the vision-language anchored adaptation modules.
>
> To directly address this concern, we have conducted **additional robustness tests in cluttered environments** for three tasks (including *pouring*, *inserting*, and *reorient*) using a dual-arm humanoid robot (see new `Fig. 8` in the Appendix). In each scenario:
>
> > (1) We introduced **≥5 distractor objects** with varying categories, shapes, or sizes.
>
> > (2) Distractors were selected to produce **semantic ambiguity**, **spatial occlusions**, and **potential collision interferences**, challenging both perception and execution.
>
> > (3) Tasks were specified only by natural-language instructions, requiring the VLM to **correctly localize the intended target** among distractors.
>
> > (4) During execution, we applied **dynamic perturbations** such as partial occlusion or object displacement to test persistent grounding stability.
>
> Finally, **VLBiMan successfully completed these tasks with high success rates** comparable to those reported in `Table 1` and `Table 2` of the main paper. Results are summarized in `Appendix C.8`, and all corresponding videos (including `new_exps_clutter_test.mp4`) have been added to the supplementary materials. We hope these new experiments directly alleviate concerns regarding VLBiMan's robustness under clutter.
>
> ***
>
> ## Q1: Human efforts required in spatiotemporal segmentation
>
> Thank you for raising this question. In VLBiMan's design, human-in-the-loop refinement is only used during Stage 1: spatiotemporal decomposition of the single demonstration, and the required effort is minimal. In practice, this refinement is only triggered when we need to ensure certain **safety-critical or precision-critical adjustments** for the extracted actions or waypoints. Examples include: **(1)** Ensuring a **stable and safe grasp** for the mug handle (a slightly backward-tilted pose rather than a naive top-down grasp). **(2)** Fixing the wrist orientation during **counterclockwise bottle opening**, avoiding deformation or torque misalignment. Such adjustments are **rare and lightweight**, and the operator does not need to be an expert.
>
> Importantly: **(1)** Once Stage 1 is completed, its cost **does not repeat** for future executions. **(2)** Stage 2 (VLM-based Adaptation) and Stage 3 (Trajectory Composition) fully automate perception and adaptation thereafter. **(3)** Only when new skills are composed to form brand-new multi-step tasks does Stage 1 need to be revisited. For transparency, we have added further discussion to `Appendix C.6`, explaining the human-related refinement logic and its limited impact on scalability.
>
> Future improvements aim toward **fully automatic decomposition**, possibly leveraging glove-based capture (DexCap) or handheld tools (UMI/DexUMI), though these introduce new trade-offs such as morphology mismatch or motion remapping errors.

---

> ### Author Response · Authors · 2025-11-26
> **Response by Authors to Reviewer uWbN [Part 2]**
>
> ## Q2: Why not use off-the-shelf 6D pose estimation?
>
> We appreciate this insightful question. The choice to avoid 6D pose estimation in favor of a lightweight mask-based orientation estimation was deliberate, for four key reasons:
>
> > ### 1. Efficiency and CAD model requirements
> Even state-of-the-art 6D pose estimators (e.g., FoundationPose) require **object CAD models**. Without them, performance degrades significantly. Moreover, inference costs can approach **~1s per frame on consumer GPUs**, making them impractical for responsive bimanual control. In contrast, our mask-based approach is **real-time, CAD-free**, and better suited for fast adaptation tasks.
>
> > ### 2. Limited category generalization
> 6D pose estimators only support objects seen during training (bottles, mugs, bowls, etc.). In VLBiMan, however, we must handle **unseen categories** such as: *marker pens*, *spatulas*, *scoops*, and *funnels*. Existing methods fail or generalize poorly to these object families.
>
> > ### 3. Symmetry-induced ambiguities
> 6D pose estimation inherently struggles with objects that have **rotational or reflective symmetry**, often producing unstable or multi-modal orientation outputs. Objects used in VLBiMan (e.g., *bottles*, *mugs*, *bowls*) are strongly symmetrical. Mask-based orientation reasoning avoids these issues entirely.
>
> > ### 4. Task relevance: full 6D pose is unnecessary
> VLBiMan **does not require** estimation of arbitrary object states. For many tasks: bottles in *pouring* are always upright; bowls in *tool-use scoop* are never inverted. The orientation estimation is needed only for specific tools (marker, spatula, lying down bottles). Unexpected or abnormal states are treated as failure or exception cases, consistent with the broader robot manipulation literature.
>
> We added a dedicated explanation in `Appendix B.1` (red-highlighted) to further clarify this design choice.
>
> ## Q3: Handling potential collisions during motion
>
> Thank you for highlighting this key concern. VLBiMan currently addresses collision risks in two primary way:
>
> > ### 1. Pre-grasp collision avoidance
> Our orientation estimation and synthesis strategy (refer `Section 3.3`), combined with near-target compensation, automatically adjusts approach trajectories to **avoid premature end-effector collisions**.
>
> > ### 2. Post-grasp bimanual coordination safety
> During Stage 1 (spatiotemporal decomposition), we analyze the one-shot demonstration to detect moments where the **two arms move too closely**, potentially causing collision during collaborative motion. If identified, we will adjust waypoints, slightly reshape the primitives, and verify the correction during human-in-the-loop refinement. This ensures that **all extracted primitives are intrinsically safe** for reproduction.
>
> For significantly more complex scenarios (e.g., large discrepancies in object dimensions or dynamically changing obstacles), a dedicated real-time motion planning module (e.g., cuRobo) would indeed be needed. Such advanced dynamic avoidance is not the focus of this paper but represents an exciting direction for improving VLBiMan in future work.
>
> ***
>
> ## Summary and Closing Remarks
>
> Once again, we sincerely thank the reviewer for the careful reading, constructive insights, and positive assessment of our work. Your recognition of VLBiMan's conceptual contribution, data efficiency, and system robustness is deeply appreciated. We have expanded our supplementary materials with new cluttered-scene evaluations, detailed discussions on spatiotemporal refinement, extended analysis on why 6D pose estimation is unsuitable for our setting, and clarifications regarding collision handling.
>
> We hope that these additions meaningfully address your concerns and help further demonstrate the significance, practicality, and novelty of VLBiMan.

---

> > ### Comment · Reviewer_uWbN · 2025-11-27
> >
> > Thank you for your detailed replies! Since all my concerns have been resolved, I decide to raise my score to 8.

---

> > > ### Author Response · Authors · 2025-11-28
> > > **Reply to Reviewer uWbN**
> > >
> > > Thank you so much for your prompt reply and improved rating. We are happy to have resolved your concerns.

---

### Official Review · Reviewer_nQo1 · 2025-11-05

**Soundness:** 3
**Presentation:** 3
**Contribution:** 2
**Rating:** 4
**Confidence:** 3

**Summary:**

This paper presents VLBiMan, a framework for bimanual robotic manipulation that learns from a single human demonstration and adapts to new objects and scenes using VLMs for semantic and geometric anchoring. The system decomposes demonstrations into reusable motion primitives, enabling generalization across tasks, object categories, and robot embodiments. The method is demonstrated on ten real-world tasks.

**Strengths:**

- The approach is demonstrated on ten real-world tasks.
- The method shows improvements over recent baselines, like ReKep.
- The supplementary shows the cross-embodiment transferability of the proposed approach.

**Weaknesses:**

- The algorithm requires human-in-the-loop refinement for the spatio-temporal segmentation. This aspect is mentioned only briefly (Line 237) but should be stated more explicitly, as it directly affects the level of automation of the pipeline. It is also unclear whether this refinement requires expert intervention.
- The reliance on mask centroids and contact points is a simplistic heuristic that may fail under occlusion or complex object geometries. This limitation requires a more detailed discussion of failure cases.
- The dynamic collision compensation introduces several design choices that appear brittle and may not generalize well.

**Questions:**

- How much manual intervention is needed during the human-in-the-loop segmentation refinement, and could this step be automated?
- How does the system handle ambiguous or incorrect segmentation results from the vision-language models, especially in cluttered scenes?

---

> ### Author Response · Authors · 2025-11-26
> **Response by Authors to Reviewer nQo1 [Part 1]**
>
> ## General Response
>
> We sincerely thank the reviewer for the time and effort spent assessing our submission, as well as for the constructive feedback provided. We are grateful for the positive recognition of our real-world validation across ten diverse tasks, our improvements over strong recent baselines such as ReKep, and the supplementary demonstration of cross-embodiment transferability. We particularly appreciate the reviewer's acknowledgement that the method is technically sound and clearly presented.
>
> Our goal with VLBiMan is to take a step toward generalizable bimanual manipulation from minimal human supervision, by combining **(1)** task-aware motion decomposition from one human demonstration, **(2)** vision-language grounded anchoring for adaptation to new objects and scenes, and **(3)** a unified hybrid-control policy that preserves human-like inter-arm coordination. We are glad that the reviewer recognized several of these strengths in the current submission.
>
> At the same time, we appreciate the reviewer's thoughtful concerns regarding the human-in-the-loop refinement, the simplicity of certain heuristics (e.g., centroid-based anchor extraction), and the design choices involved in dynamic collision compensation. These are important aspects, and we thank the reviewer for highlighting them. In the revised version, we will make these design trade-offs more explicit, provide expanded discussions of limitations and failure modes, and clarify where expert intervention is (and is not) required. Below, we address each of the raised weaknesses and questions in detail.
>
> ***
>
> ## W1 & Q1: On human-in-the-loop refinement during spatio-temporal decomposition
>
> Thank you for raising this important point. We agree that the spatio-temporal segmentation in Stage-1 of VLBiMan may occasionally require light human-in-the-loop refinement, primarily to verify or adjust a small number of motion waypoints when the one-shot bimanual demonstration contains suboptimal poses. Typical cases include ensuring a stable angled grasp on a mug handle (rather than a flat or top-down grasp) in the *pouring* task or enforcing a strictly vertical orientation of the right arm during untwisting to avoid excessive lateral forces on the bottle in the *unscrew* task.
>
> Importantly, these refinements are rare across the ten tasks and **do not require** expert-level skill. Once the Stage-1 decomposition is completed for a task, the system **never re-executes** this step unless a new composite task (e.g., *unscrew+pouring*) is created. All subsequent variations (including new objects, poses, scene layouts, and clutter) are handled fully automatically through Stage-2 VLM-based adaptation and Stage-3 trajectory composition. Thus, the limited human involvement does not hinder the scalability of VLBiMan.
>
> We agree that pursuing fully automated segmentation is an important direction. Motion capture gloves (e.g., DexCap) or human-operated teleoperation devices (e.g., UMI / DexUMI) are promising but still introduce cross-embodiment mapping errors, DOF misalignment, and calibration overhead. We discuss these trade-offs, along with the scope of human involvement, in `Appendix C.6`.
>
> ## W2: On the simplicity and robustness of centroid/contact-based object anchoring
>
> We appreciate the reviewer's insightful comments. As noted, VLBiMan uses two types of object representative points (`Fig. 3`): **(1)** mask centroids, and **(2)** the lowest contact point with the table. This abstraction was intentionally chosen for its generality and simplicity, and it is validated across all ten tasks.
>
> To further evaluate its robustness under occlusion and complex geometries, we conducted additional experiments on the *inserting* task, which uses both types of representative points (a long slender object represented by a centroid, and a container represented by a table contact point). The original used objects in the *inserting* task are a marker pen and a mug cup. We here expanded testing to many novel elongated objects (e.g., spoons, brushes, spatulas, syringes) and to cups/mugs with highly varied shapes and rims.
>
> Even though occasional failure cases appear (as expected with any simplified abstraction), the overall performance remains strong, demonstrating surprisingly high robustness to partial occlusions and geometric variability. These extended results are included in `Appendix C.7`, with videos in the supplementary material (`new_exps_rep_pts_test.mp4`).

---

> ### Author Response · Authors · 2025-11-26
> **Response by Authors to Reviewer nQo1 [Part 2]**
>
> ## W3 & Q2: On the dynamic collision compensation mechanism and handling ambiguous VLM segmentations
>
> We fully understand the reviewer's concerns regarding the simplicity of our dynamic collision compensation design. Indeed, while sophisticated real-time planning systems such as **cuRobo** could be incorporated, our goal in this work is not to optimize low-level collision avoidance, but to study how an one-shot bimanual demonstration can be decomposed, adapted, and recomposed for generalizable manipulation across diverse objects and tasks.
>
> Thus, we deliberately focused on a lightweight approach that avoids premature collisions during pre-grasp and dual-arm alignment phases while preserving the high-level motion semantics extracted from demonstrations. Despite this minimalist design, VLBiMan generalizes surprisingly well. To further evaluate the claimed robustness, we tested complex cluttered scenes in *pouring/inserting/reorient* tasks, requiring VLBiMan to: **(1)** disambiguate VLM segmentations in the presence of distractors, **(2)** avoid unintended collisions with irrelevant objects, and **(3)** operate reliably under partial occlusion of the target objects.
>
> Across these settings, the system maintained strong performance, indicating that our simple yet efficient collision compensation module generalizes adequately to challenging real-world conditions. Results are added and reported in `Appendix C.8`, with corresponding execution videos (`new_exps_clutter_test.mp4`).
>
> ***
>
> ## Summary and Closing Remarks
>
> We sincerely thank the reviewer again for the thoughtful evaluation and constructive feedback. Your comments helped us clarify the scope of VLBiMan, strengthen the discussion of its practical limitations, and provide additional experiments that deepen the technical understanding of our approach. We believe the expanded analyses on **(1)** human-in-the-loop segmentation refinement, **(2)** robustness of centroid/contact-point anchoring, and **(3)** cluttered-scene collision handling, together with the supplementary experiments in `Appendices C.6–C.8`, substantially improve the clarity and completeness of the work. We appreciate your recognition of our system's real-world performance, cross-embodiment generalization, and advantages over strong baselines. We hope that our responses have addressed your concerns and conveyed the core methodological value of VLBiMan. Thank you again for your time, effort, and constructive contributions to improving this paper.

---

### Author Response · Authors · 2025-11-26
**Author Final Remarks**

We sincerely thank all reviewers for their thoughtful feedback, constructive suggestions, and encouraging evaluations of our work. Across the four reports, reviewers consistently highlighted VLBiMan's conceptual clarity, its intuitive modular pipeline, strong empirical performance, and the demonstrated ability to generalize across new objects, poses, and embodiments. We greatly appreciate these positive assessments and have carefully addressed all concerns through clarification, additional experiments, and expanded discussions in the appendix.

***

**1. Addressing concerns about novelty and modular design.**
Several reviewers noted that VLBiMan's components are individually standard. We clarified that the contribution lies not in reinventing isolated modules but in proposing a vision-language–anchored decomposition–adaptation–composition pipeline tailored for one-shot generalizable bimanual manipulation. The method requires no training or retraining, operates fully geometry- and VLM-grounded, and uniquely supports cross-object, cross-pose, and cross-embodiment transfer—capabilities that prior modular or monolithic VLA systems do not jointly achieve.

**2. On human-in-the-loop refinement during segmentation.**
Following reviewer comments, we added a new subsection in `Appendix C.6` detailing how human refinement is only needed once per demonstration, and never required at test time. We further show that our hierarchical segmentation is robust even with imperfect boundaries, and that refinement ensures better temporal alignment for complex dual-arm tasks.

**3. On the choice of using 2D mask–based object representing points and principal axes (instead of 6D pose estimation).**
Reviewers questioned why we do not use off-the-shelf 6D pose estimators. In `Appendix B.1`, we added discussion showing that (1) many household objects in our tasks are texture-poor, symmetric, or reflective—settings where 6D methods can be unstable; (2) our tasks only require orientation around a dominant axis and a consistent representing point, not a full SE(3) pose; and (3) our mask-derived centroids and front-edge contacts generalize extremely well across object shapes. We additionally conducted new experiments on the *inserting* task with novel objects, confirming that this simple representation remains robust and reliable (refer `Appendix C.7`).

**4. On robustness and clutter.**
Reviewers asked about stability under real-world disturbances. In response, we expanded `Appendix C.8` with new experiments under cluttered scenes and external perturbations. Results show that VLBiMan maintains performance even when objects are moved or reoriented mid-execution, demonstrating strong resilience enabled by continuous geometric anchoring.

**5. On interpolation density in pre-grasp motion.**
A new ablation study (`Appendix C.9`) investigates the number of interpolation points used in the pre-grasp phase. We show that denser interpolation significantly reduces premature end-effector contact, produces smoother trajectories, and enables tolerance to intentional object disturbances—further validating the design choice.

**6. On comparison to closed-loop VLA models.**
To clarify reviewer concerns, we added additional discussion explaining that current VLA policies are either (1) not dual-arm; (2) not capable of long-horizon multi-stage manipulation; or (3) require large training datasets and lack one-shot adaptability. Thus, fair direct comparisons are currently infeasible. Nonetheless, we emphasize that VLBiMan is complementary: it can serve as a structured supervision source or policy scaffolding for future closed-loop VLA systems.

**7. On computational efficiency.**
We now provide timing measurements for each module. All components (including image-moment orientation extraction) together run in about 1s per loop, confirming that the system is lightweight and suitable for deployment.

**8. On cross-embodiment transferability.**
Reviewers found the embodiment transfer results compelling. To further strengthen this point, we added new real-world evaluations on a dual-arm humanoid platform across four tasks (inserting, unscrew, pouring, reorient). Both quantitative results and qualitative executions (`Sec. C`) show that VLBiMan achieves success rates comparable to the original platform, demonstrating its strong generalization across robot morphologies.

***

Overall, we have substantially expanded the appendix (four new subsections), added new experiments (cluttered scenes, interpolation density ablation, robustness of representing points, cross-embodiment tests), and clarified all technical questions raised by reviewers. We believe these additions further strengthen the significance, completeness, and reproducibility of VLBiMan. We are grateful for the reviewers' time and enthusiasm, and we hope that our revisions will address all remaining concerns.

---

### Meta-Review · Area_Chair_nzZC · 2026-01-07

**Summary:**

The paper got mixed ratings, 4, 6, 4, 8.

Reviewers commonly recognize the strengths of the paper in data efficiency and generalization, achieved by learning reusable skills from a single demonstration. They also note the practical efficacy of the approach and the real-world evaluations conducted across 10 tasks.

The reviewers’ concerns focus on insufficient descriptions of certain aspects of the paper, including the human-in-the-loop refinement process, hyperparameters, and sensitivity to segmentation quality. The reviewers’ feedback is diverse and is well summarized in the authors’ final response.

**Reviewer Concerns:**

The AC finds that the authors’ responses address most of the reviewers’ concerns.

There are no widely shared concerns among the reviewers. The authors’ responses to individual reviewers and the final summary convincingly address the majority of the raised issues. In particular, the authors address missing methodological details and provide several additional experiments, including evaluations in cluttered scenes, a new insertion task with novel objects, and further ablation studies.

**Reviewer Scores:**

The AC expects that the positive ratings from reviewers remain unchanged and that the negative ratings may increase.

The authors provide convincing evidence to support the strengths of their method and respond effectively to the concerns raised by the reviewers. The paper contains clear novelty, as commonly recognized by all reviewers.

Based on these considerations, the AC recommends acceptance of this paper.

---

### Decision · Program_Chairs · 2026-01-26

Accept (Poster)